# GRAPH CONVOLUTIONS ENRICH THE SELF-ATTENTION IN TRANSFORMERS!

## ABSTRACT

Transformers, renowned for their self-attention mechanism, have achieved the state-of-the-art performance across various tasks in natural language processing, computer vision, time-series modeling, etc. However, one of the challenges with deep Transformer models is the oversmoothing problem, where representations across layers converge to indistinguishable values, leading to significant performance degradation. We interpret the original self-attention as a simple graph filter and redesign it from a graph signal processing (GSP) perspective. We propose graph-filter-based self-attention (GFSA) to learn a general yet effective one, whose complexity, however, is slightly larger than that of the original self-attention mechanism. We demonstrate that GFSA improves the performance of Transformers in various fields, including computer vision, natural language processing, graph pattern classification, speech recognition, and code classification.

## 1 INTRODUCTION

Transformers are arguably one of the best feats in the field of deep learning. They are now showing the state-of-the-art performance in various fields, ranging from computer vision to natural language processing to graph pattern classification to speech recognition, and so forth (Vaswani et al., 2017; Devlin et al., 2019; Radford et al., 2018; 2019; Dosovitskiy et al., 2021; Touvron et al., 2021a; Zhou et al., 2021a; Liu et al., 2021; Gulati et al., 2020; Latif et al., 2023; Ying et al., 2021; Rampášek et al., 2022). Recently, there have been several studies conducted on better understanding them (Gong et al., 2021; Ali et al., 2023; Wang et al., 2022); there exists a common agreement among researchers that the self-attention is one of the keys leading to the success.

However, there also exist several studies pointing out potential limitations of the self-attention (Zhou et al., 2021a; Dong et al., 2021; Gong et al., 2021; Guo et al., 2023). For instance, Shi et al. (2022) revealed an analogy between the self-attention and the residual graph convolutional network (GCN), showing that BERT also suffers from a notorious problem of GCNs, called *oversmoothing*, i.e., tokens' latent representations become similar to each other at the deeper layers. In every self-attention layer, value vectors are aggregated in a weighted average manner since each row-wise sum of the attention matrix is always 1. Although each self-attention layer has its own attention matrix, this aggregation method causes the oversmoothing problem, not only in Transformers but also in graph neural networks (Oono & Suzuki, 2020; Cai & Wang, 2020; Wang et al., 2021a; Rusch et al., 2023; Keriven, 2022; Zhou et al., 2021a; Gong et al., 2021; Yan et al., 2022; Noci et al., 2022; Shi et al., 2022; Wang et al., 2022; Ali et al., 2023; Wu et al., 2023a;b). However, we confine our discussion to the oversmoothing of Transformers (cf. Section 2).

Being inspired by them, we redesign the self-attention from the perspective of graph signal processing (GSP). However, performing graph convolutions in the self-attention layer may incur non-trivial computational overheads. Therefore, our key design point is to learn a general but effective graph filter with minimal overheads. In general, a graph filter on a graph $\mathcal{G}$ is represented by a polynomial expression based on its adjacency or Laplacian matrix — in this regard, the existing self-attention mechanism can be understood as the simplest graph filter with $\bar{A}$ only, where $\bar{A} \in [0,1]^{n \times n}$ means a learned attention matrix and $n$ is the number of input tokens.

Our proposed graph filter consists of an identity term and two matrix polynomial terms, $\bar{A}$ and $\bar{A}^K$ — one can design better filters with more polynomial terms but we avoid it since Transformers already require very large computation. The $K$-th power, $\bar{A}^K$, may also require high computation

when the number of tokens is large. To avoid this, we further approximate $\bar{\boldsymbol{A}}^K$ using the element-wise first-order Taylor approximation. Therefore, one can consider that our proposed graph filter is the very next complicated filter after the one used by the original self-attention mechanism. However, its efficacy is tremendous for various Transformers in various fields (cf. Figure 1).

Our proposed filter enriches the self-attention with more diverse frequency information (see Figure 2(a)) — low (resp. high) frequency signals on $\mathcal{G}$ mean neighboring nodes have similar (resp. different) values. Therefore, our method is able to not only effectively address the oversmoothing problem but also learn better latent representations for downstream tasks.

There exist a couple of prior works to enrich the self-attention mechanism with high frequency information (Wang et al., 2022; Bai et al., 2022). In comparison with them, our proposed graph filter is distinctive in the following aspects: i) our proposed filter is more effective and shows better performance with comparable computational overheads, ii) our proposed filter is well-aligned with recent advancements in the GCN community — in other words, some graph filters used by recent advanced GCN methods are special cases of our proposed graph filter, which is not the case for prior works, and iii) other methods were typically studied for certain domains only whereas we test our method in 6 domains — for instance, DiversePatch (Gong et al., 2021) works only for Vision Transformers.

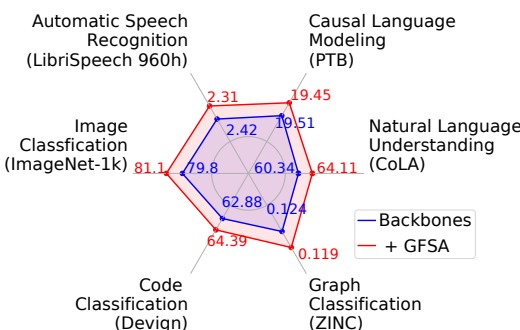

Figure 1: Our proposed GFSA performs better than selected Transformers in various domains. We achieve these results with only tens to hundreds of additional parameters to Transformers.

We replace the self-attention layer of selected Transformers in various fields with our proposed graph filter-based layer without changing other parts. Therefore, the accuracy increases in them are solely by our proposed graph filter-based self-attention. These enriched Transformers increase the model performance by 1.63% for image classification, 6.25% for natural language understanding, 0.31% for causal language modeling, 4.03% for graph classification, 4.76% for speech recognition, and 2.40% for code classification (see Figure 1). We summarize our core contributions as follows:

- We provide a novel perspective on self-attention as a graph filter. This perspective allows us to design more effective self-attention that can address the oversmoothing problem.
- We propose a graph filter-based self-attention (GFSA) mechanism that is general yet effective. GFSA learns a graph filter with an identity term and two polynomial terms, which is more effective than the simple self-attention mechanism.
- We demonstrate that GFSA can significantly improve the performance of Transformers on a variety of tasks. GFSA achieves improved results on computer vision, natural language processing, graph pattern classification, speech recognition, and code classification.

## 2 BACKGROUND & RELATED WORK

### 2.1 SELF-ATTENTION IN TRANSFORMERS

The core building block of the Transformer architecture is the self-attention mechanism, which enables the model to learn attention patterns over its input tokens (Vaswani et al., 2017). The self-attention mechanism, denoted as SA : $\mathbb{R}^{n \times d} \to \mathbb{R}^{n \times d}$, can be expressed as follows:

$$\text{SA}(\boldsymbol{X}) = \text{softmax}\left(\frac{\boldsymbol{X}\boldsymbol{W}_{\text{key}}(\boldsymbol{X}\boldsymbol{W}_{\text{qry}})^{\intercal}}{\sqrt{d}}\right)\boldsymbol{X}\boldsymbol{W}_{\text{val}} = \bar{\boldsymbol{A}}\boldsymbol{X}\boldsymbol{W}_{\text{val}}, \tag{1}$$

where $\boldsymbol{X} \in \mathbb{R}^{n \times d}$ is the input feature matrix, $\boldsymbol{W}_{\text{key}} \in \mathbb{R}^{d \times d}$, $\boldsymbol{W}_{\text{qry}} \in \mathbb{R}^{d \times d}$, and $\boldsymbol{W}_{\text{val}} \in \mathbb{R}^{d \times d}$ are the key, query, and value trainable parameters, respectively, and $d$ is the dimension of each token. The self-attention mechanism allows the model to weigh the importance of each token in the input sequence relative to the others, enabling the model to capture long-range contextual information

better. The Transformer architecture includes multiple layers, each with a multi-head self-attention layer followed by a position-wise feed-forward layer.

## 2.2 SELF-ATTENTION AND GRAPH CONVOLUTIONAL FILTER

The self-attention matrix used in Transformers can be seen as a symmetrically normalized adjacency matrix of a graph (Shi et al., 2022; Guo et al., 2023; Maskey et al., 2023). A weighted graph $\mathcal{G}$ with adjacency matrix $\boldsymbol{A}$ can be constructed by using the input tokens as $n$ nodes and the edge weights between node $i$ and node $j$ as $\exp((\boldsymbol{X}\boldsymbol{W}_{\text{qry}})^{\intercal}(\boldsymbol{X}\boldsymbol{W}_{\text{key}}))$. We can rewrite the self-attention matrix $\bar{\boldsymbol{A}}$ as $\frac{\exp(\boldsymbol{X}\boldsymbol{W}_{\text{qry}})_i^{\intercal}(\boldsymbol{X}\boldsymbol{W}_{\text{key}})_j}{\sum_{k=1}^{d}\exp(\boldsymbol{X}\boldsymbol{W}_{\text{qry}})_i^{\intercal}(\boldsymbol{X}\boldsymbol{W}_{\text{key}})_k}$. This allows $\bar{\boldsymbol{A}}$ to be interpreted as the symmetrically normalized adjacency matrix. In other words, $\bar{\boldsymbol{A}} = \boldsymbol{D}^{-1}\boldsymbol{A}$, where $\boldsymbol{D} = \text{diag}(d_1, d_2, \ldots, d_n)$ and $d_i = \sum_j \boldsymbol{A}_{i,j}$.

Our new attention method is designed on top of graph signal processing (GSP) which has a close connection to discrete signal processing (DSP) (Sandryhaila & Moura, 2013; 2014). In DSP, a discrete signal with a length of $n$ can be represented by a vector $\boldsymbol{x} \in \mathbb{R}^n$. Let $\boldsymbol{g} \in \mathbb{R}^n$ be a filter that we want to apply to $\boldsymbol{x}$. The convolution $\boldsymbol{x} * \boldsymbol{g}$ can be written as follows:

$$\boldsymbol{y}_i = \sum_{j=1}^{n} \boldsymbol{x}_j \boldsymbol{g}_{i-j}, \tag{2}$$

where the index, denoted as $i$, refers to the $i$-th element in each vector.

GSP can be understood as a generalized concept of DSP. Signals are defined on the nodes of a graph, and the graph's structure influences signal processing operations. In addition, the graph convolution filter with $n$ nodes can be written with a shift operator $\boldsymbol{S}$ as follows:

$$\boldsymbol{y} = \sum_{k=0}^{K} w_k \boldsymbol{S}^k \boldsymbol{x} = \boldsymbol{H}\boldsymbol{x}, \tag{3}$$

where $\boldsymbol{x} \in \mathbb{R}^n$ is a 1-dimensional graph signal, $K$ is the maximum order of polynomial, and $w_k \in [-\infty, \infty]$ is a coefficient. $\boldsymbol{S}$ is an $n \times n$ matrix where $(i, j)$-th element is non-zero if and only if there is an edge from node $i$ to $j$. Two representative samples of $\boldsymbol{S}$ are adjacency and Laplacian matrices. The linear and shift-invariant graph filter $\boldsymbol{H}$ is the same as $\sum_{k=0}^{K} w_k \boldsymbol{S}^k$ with a large enough value of $K$, which is called *matrix polynomial* (Marques et al., 2020). We note that this graph filtering operation can be extended to $d$-dimensional cases as in Equation 1. Being inspired by (Zou et al., 2022; Maskey et al., 2023), we rely on the singular value domain analysis to understand the low/high-pass filter characteristics of filters (cf. Fig. 2). See more discussion in Appendix N.

In the context of self-attention within Transformers, the core part of the self-attention in Equation 1, i.e., $\bar{\boldsymbol{A}}\boldsymbol{X}$, can be considered as a $d$-dimensional graph filter with $\bar{\boldsymbol{A}}$ only, where $\boldsymbol{H} = \bar{\boldsymbol{A}}$. Our goal in this paper is design a simple (for computational purposes) yet effective form of $\boldsymbol{H}$.

## 2.3 OVERSMOOTHING IN GCNS AND TRANSFORMERS

Oversmoothing is a phenomenon observed in deep learning models, especially in GCNs (Kipf & Welling, 2017; Veličković et al., 2018). As information is aggregated over multiple layers for multiple nodes (tokens), latent representations tend to become similar to each other, leading to a loss of distinctiveness in the representations (Oono & Suzuki, 2020; Zhou et al., 2020; Rusch et al., 2023).

Surprisingly, an oversmoothing-like phenomenon is also observed in Transformers (Wang et al., 2022; Shi et al., 2022). Unlike CNNs, Transformers can not benefit from simply deepening layers after a certain depth. Earlier studies hypothesize that this may be due to issues such as the attention or feature collapse or due to uniformity among patches or tokens (Zhou et al., 2021a; Gong et al., 2021; Yan et al., 2022). Dong et al. (2021) also point out that the output of a pure Transformer, i.e., an attention mechanism without skip connections or MLPs, tends to converge to a rank-1 matrix (Dong et al., 2021). This analysis is followed by (Noci et al., 2022), which suggests that rank collapses incur vanishing gradients of attention queries and keys.

In this context, the self-attention acts as a low-pass filter, since the self attention calculates the weighted average of the value vectors of tokens. Wang et al. (2022, Theorem 1) also reveal that the

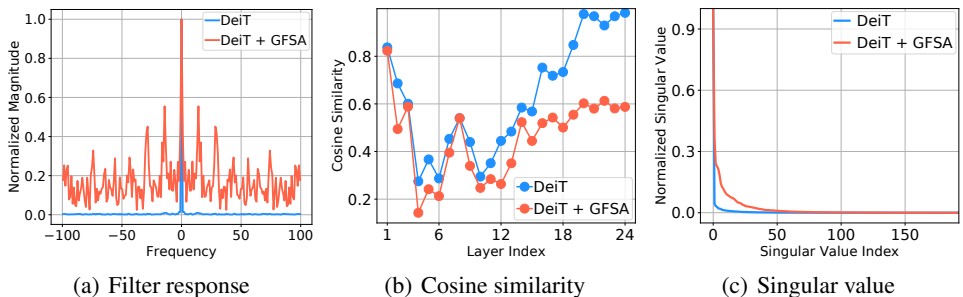

(a) Filter response  (b) Cosine similarity  (c) Singular value

Figure 2: Filter frequency response, cosine similarity, and singular values on ImageNet-1k for DeiT-S and DeiT-S + GFSA. Details and more visualizations are in Appendix D.

self-attention is a low-pass filter, continuously reducing high-frequency information. This nature contributes to the oversmoothing phenomenon as unique high-frequency features are lost in deeper layers of the network, further worsening the uniformity of token representations. Therefore, we extend the term "oversmoothing" to describe the degeneration challenge observed in Transformers.

There have been proposed many empirical countermeasures for Vision Transformer (ViT), such as patch diversification (Zhou et al., 2021b; Gong et al., 2021), rank collapse alleviation (Zhou et al., 2021a; Zhang et al., 2021), and training stabilization (Touvron et al., 2021a; Zhai et al., 2023). Similar alleviating methods have been also proposed in the field of NLP, such as unsupervised learning (Chen et al., 2023), and resolve the oversmoothing and the token uniformity (or information diffusion) problems (Dong et al., 2021; Yan et al., 2022). There are studies on utilizing high frequency information via frequency domain analyses (Wang et al., 2022; Bai et al., 2022), but they are not designed on top of graph filtering perspectives. This paper addresses the oversmoothing problem with graph filters since the self-attention mechanism is a basic graph filtering operation as seen in the previous subsection.

## 3  PROPOSED METHOD

### 3.1  GRAPH FILTER-BASED SELF-ATTENTION LAYERS

Let $\bar{\boldsymbol{A}} \in [0, 1]^{n \times n}$, where $n$ is the number of tokens in the input to the self-attention layer, be a self-attention matrix. Since transformers use multi-head self-attentions, there are multiple such matrices. For simplicity but without loss of generality, we discuss only one head in one layer.

In the perspective of GSP, $\bar{\boldsymbol{A}}$ corresponds to a special case where i) nodes are completely connected, and ii) $\bar{\boldsymbol{A}}$ is normalized. $\bar{\boldsymbol{A}}$ can be used as a shift operator. When $K$ is large, however, it is hard to calculate $\bar{\boldsymbol{A}}^K$. We use the following element-wise first-order Taylor approximation:

$$\bar{\boldsymbol{A}}^K \approx \bar{\boldsymbol{A}} + (K-1)(\bar{\boldsymbol{A}}^2 - \bar{\boldsymbol{A}}), \tag{4}$$

where $\bar{\boldsymbol{A}}^2 - \bar{\boldsymbol{A}}$ is an approximated derivative at the position of $K = 1$ (see details in Appendix P). At the end, we propose to use the following graph filter where the two lowest-order terms and one high-order term of the matrix polynomial are used:

$$\boldsymbol{H} := w_0 \boldsymbol{I} + w_1 \bar{\boldsymbol{A}} + w_K (\bar{\boldsymbol{A}} + (K-1)(\bar{\boldsymbol{A}}^2 - \bar{\boldsymbol{A}})), \tag{5}$$

where $w_0$, $w_1$, $w_K$ are coefficients and $K$ is a hyper-parameter. The coefficients can be learnable weights and we learn them with gradient descent algorithms.

Our proposed graph filter-based self-attention (GFSA) is defined with the graph filter $\boldsymbol{H}$ as follows:

$$\text{GFSA}(\boldsymbol{X}) = \boldsymbol{H}\boldsymbol{X}\boldsymbol{W}_{\text{val}}. \tag{6}$$

We replace the original self-attention layer in various Transformers with the proposed graph filter-based layer without changing other parts. Therefore, GFSA can be plugged into any Transformers that rely on the self-attention.

## 3.2 THEORETICAL ANALYSIS

In this subsection, we provide a theorem that provides an upper bound on the error introduced by approximating the power of a matrix, specifically using the first-order Taylor expansion. The theorem specifically analyzes the error for the matrix $\bar{A}^K$ when approximated using a first-order Taylor expansion.

**Theorem 3.1** (Error bound for first-order Taylor approximation). *Define the error term as the difference between the exact value and the first-order Taylor approximation of $\bar{A}^K$ :*

$$E_K = ||\bar{A}^K - (\bar{A} + (K-1)(\bar{A}^2 - \bar{A}))||, \tag{7}$$

*where $|| \cdot ||$ denotes the $L_\infty$ norm. Then, the error $E_K$ can be bounded as follows:*

$$E_K \leq 2K. \tag{8}$$

The error bound provides an upper limit for the difference between the true value of $\bar{A}^K$ and its approximation using the given formula. When the row-sum of $\bar{A}$ is 1 due to softmax. The proof of Theorem. 3.1 is in Appendix A.

Computing high-order matrix powers can be computationally intensive, especially for large matrices such as the self-attention in natural language processing where the number of tokens (words) can be tens of thousands. The first-order Taylor approximation provides a simpler computation that can significantly reduce the required computational resources and time. Theorem. 3.1 ensures that even with this simplification, the error can be bounded and is predictable.

## 3.3 DISCUSSION

This subsection provides a discussion on GFSA in terms of the comparison with other models and the mitigation of the oversmoothing problem.

**Comparison to Transformers**  In the field of computer vision, there have been recent researches on adjusting the frequency response of ViT. HAT (Bai et al., 2022) improves ViT by directly increasing the high-frequency components of images through adversarial training, i.e., modifying inputs. Wang et al. (2022) suggest that they re-balance the low and high-frequency components of the attention map and feature map, respectively (Wang et al., 2022). In the NLP domain, Shi et al. (2022) redesign BERT with multiple fusions in (Shi et al., 2022), inspired by JKNet (Xu et al., 2018).

**Comparison to GCNs**  Comparisons to GCNs that can be interpreted as graph filters (Kipf & Welling, 2017; Defferrard et al., 2016; Gasteiger et al., 2019) are inevitable. GFSA without a high-order term is analogous to ChebNet (Defferrard et al., 2016) with $K = 1$. In addition, GFSA reduces to the vanilla GCN (Kipf & Welling, 2017) when $K = 1$, $w_0 = 0$, $w_1 = 1$. GPR-GNN (Chien et al., 2021), which approximates graph convolutions using the monomial basis, is identical to GFSA if our GFSA only considers up to an order of 1 and learns coefficients. When we use only a high-order term and $w_K$ is learned to negative value, GFSA can become similar to the reaction-diffusion layer of GREAD (Choi et al., 2023), $\bar{A}X + \beta(\bar{A} - \bar{A}^2)$, depending on the higher order terms.

**How to alleviate the oversmoothing problem?**  The key leading to the behaviors of our proposed filter is the coefficients $\{w_0, w_1, w_K\}$ — note that in the self-attention of Transformers, $w_0 = w_K = 0$ and $w_1 = 1$. Since our method can learn any appropriate values for them for a downstream task, it can reduce to low-pass-only, high-pass-only, or combined filters. In particular, our filter reduces to GPR-GNN and GREAD, two popular method preventing the oversmoothing for graph convolutional networks, in certain learned settings on $\{w_0, w_1, w_K\}$. We use $\bar{A}^2 - \bar{A}$ to approximate $\bar{A}^K$ (see Equation 4). As mentioned earlier, the low/high-pass filter characteristics are decided by $\{w_0, w_1, w_K\}$. Therefore, the approximated $\bar{A}^K$ by $(K-1)(\bar{A}^2 - \bar{A})$ and the coefficients $\{w_0, w_1, w_K\}$ can alleviate the oversmoothing problem of the self-attention by constituting an appropriate graph filter for a downstream task.

## 4 EXPERIMENTS

In this section, we demonstrate the effectiveness of GFSA through a series of experiments. These experiments encompass various tasks: i) language understanding and causal language modeling,

Table 1: Results comparison on validation set of GLUE tasks. **Avg** denotes the average performance on all the tasks.

| Method | #Params | CoLA | SST-2 | MRPC | QQP | STS-B | MNLI-m/mm | QNLI | RTE | **Avg** |
|---|---|---|---|---|---|---|---|---|---|---|
| BERT$_{BASE}$ (Devlin et al., 2019) | 110M | 56.79 | 93.81 | 88.70 | 88.32 | 88.16 | 84.96/84.15 | 91.63 | 66.06 | 82.51 |
| BERT$_{BASE}$ + ContraNorm | 110M | **59.89** | 93.92 | 89.88 | **88.51** | **88.36** | 85.11/84.50 | 91.84 | **69.31** | 83.48 |
| BERT$_{BASE}$ + GFSA | 110M | 59.56 | **94.15** | **90.60** | 88.46 | 88.33 | **85.12/85.06** | **91.95** | 68.95 | **83.58** |
| ALBERT$_{BASE}$ (Lan et al., 2019) | 11M | 57.86 | 92.32 | 91.80 | 85.30 | 90.37 | **85.37**/84.37 | 91.76 | 76.90 | 84.01 |
| ALBERT$_{BASE}$ + ContraNorm | 11M | 57.45 | 93.00 | **92.83** | 87.78 | 90.55 | 85.06/84.57 | **92.28** | **78.70** | 84.69 |
| ALBERT$_{BASE}$ + GFSA | 11M | **60.21** | 93.23 | 92.47 | **87.79** | 90.63 | 85.29/**84.92** | 92.17 | **78.70** | **85.05** |
| RoBERTa$_{BASE}$ (Liu et al., 2019) | 125M | 60.34 | 94.84 | 92.28 | 88.86 | 89.99 | 87.94/87.30 | 92.57 | 78.70 | 85.87 |
| RoBERTa$_{BASE}$ + ContraNorm | 125M | 63.06 | **95.41** | 93.17 | 88.91 | 90.34 | 87.88/87.40 | 92.82 | **80.51** | 86.61 |
| RoBERTa$_{BASE}$ + GFSA | 125M | **64.11** | 95.41 | **93.52** | **89.09** | **90.35** | **87.99/87.54** | **92.97** | 80.14 | **86.79** |

ii) image classification, iii) graph classification, and iv) code classification. We replace the self-attention of base Transformers in those fields with our GFSA. Our modification adds only tens to hundreds of parameters, which are negligible in comparison with the original size of base models. We summarize the runtime of base models vs. base models with GFSA in Appendix L.

## 4.1 EXPERIMENTS ON LANGUAGE MODELS

To investigate the efficacy of GFSA, we present experiments on natural language understanding and causal language modeling tasks in the subsections below.

### 4.1.1 NATURAL LANGUAGE UNDERSTANDING

**Experimental Setting.** We integrate GFSA into three pre-trained large language models: BERT, ALBERT, and RoBERTa. We evaluate them on the GLUE benchmark, which includes three categories of natural language understanding tasks: i) single-sentence tasks CoLA and SST-2; ii) similarity and paraphrasing tasks MRPC, QQP, and STS-B; iii) natural language inference tasks MNLI, QNLI, and RTE. For the MNLI task, we experiment on both the matched (MNLI-m) and mismatched (MNLI-mm) versions. Following (Devlin et al., 2019), we report Matthews correlation for CoLA, F1 scores for QQP and MRPC, Spearman correlations for STS-B, and accuracy scores for the other tasks. For each task, we select the best hyperparameters for GFSA, and the other hyperparameters are fixed. We compare our GFSA with ContraNorm (Guo et al., 2023), one of the related methods that address oversmoothing. We finetune ContraNorm with the recommended hyperparameters in the original paper. We initialize with pretrained language model and finetune with added GFSA for 5 epochs. The detailed experimental settings are in Appendix E.1.

**Results.** The results are shown in Table 1. When GFSA was plugged into backbones, average performance scores improved across all models over pure backbones. This indicates that GFSA is effective in both large models like BERT and RoBERTa, as well as relatively smaller models like ALBERT. It is worth mentioning that in the case of RoBERTa finetuned on the CoLA dataset, there is a large margin increase from 60.34% to 64.11%, which is a 3.77% improvement with only 144 additional parameters. When compared to ContraNorm, GFSA shows a greater performance improvement on average. Figure 4 in Appendix D shows that these performance enhancements can be attributed to addressing the oversmoothing issue through the designed graph filter.

### 4.1.2 CAUSAL LANGUAGE MODELING

**Experimental Setting.** We also validate the effectiveness of GFSA on causal language modeling problems. We finetune GPT2 on the following three datasets: Penn Treebank (PTB) (Marcus et al., 1993), WikiText-2 and WikiText-103 (Merity et al., 2016). Following the evaluation method in (Yao et al., 2022), we finetune models for 15 epochs with PTB, 4 epochs with WikiText-103, and 10 epochs with WikiText-2, and report the perplexity for sensitivity metric, i.e., higher perplexity means higher sensitivity. The detailed experimental settings are in Appendix F.1.

Table 2: Results comparison on GPT-2 finetuned with GFSA

| Method | #Params | PTB | WikiText-2 | WikiText-103 | Avg |
|---|---|---|---|---|---|
| GPT2 (Radford et al., 2019) | 117M | 19.513 | 20.966 | 15.939 | 18.806 |
| GPT2 + GFSA | 117M | **19.450** | **20.923** | **15.919** | **18.764** |

**Results.** Table 19 shows the perplexity on PTB, WitiText-2, and WikiText-103. Across all datasets, GPT2 with GFSA consistently outperforms the vanilla GPT2. Our GFSA improves the average perplexity from 18.806 to 18.764. Note that the performance improvements are made with only 144 additional learnable parameters for 12 layers with 12 heads, which are negligible in comparison with the size of GPT2.

## 4.2 EXPERIMENTS ON VISION TRANSFORMERS

**Experimental Setting.** We aim to demonstrate the efficacy of our GFSA across a spectrum of ViT backbones with different depth settings and training modes. We choose DeiT (Touvron et al., 2021a), CaiT (Touvron et al., 2021b), and Swin (Liu et al., 2021) as backbones, and models are trained from scratch. When training the 12-layer DeiT, we follow the same training recipe, hyperparameters, and data augmentation from (Touvron et al., 2021a). When training the 24-layer DeiT, we follow the settings of (Gong et al., 2021) and set the dropout rate to 0.2. For Cait, we apply our proposed GFSA only to the patch embedding layer. For detailed experimental settings, see Appendix G.1.

**Results.** All experimental evaluations are summarized in Table 3. We compare various state-of-the-art models on the ImageNet-1k benchmark. We select them from three classes: CNN only, CNN + Transformer, and pure Transformer. In the Transformer category, we only test with lighter models having comparable numbers of parameters, such as ViT-S and DeiT-S. The results show that the proposed GFSA successfully enhances Deit, CaiT, and Swin across all depth settings and training methods. GFSA provides additional parameters less than 72 for 12-layer DeiT while improving the top-1 accuracy by 1.63%. Compared to existing techniques, improvements by GFSA already surpasses DeepViT-24B (0.6%) (Zhou et al., 2021a), LayerScale (0.7%) (Touvron et al., 2021b), LateInsertion (0.6%) (Touvron et al., 2021b), and HAT (Bai et al., 2022) (1.38%). To sum up, we observed that both shallow and deep ViTs can achieve the following benefits from GFSA: i) The filter response plot shows GFSA can preserve higher-frequency representation (cf. Figure 2 (a)) and ii) Figure 2 (b) shows that GFSA can mitigate the increase in the cosine similarity of representation as the layer gets deeper. We also show experiments with the same ContraNorm setting in Appendix G.4.

## 4.3 EXPERIMENTS ON AUTOMATIC SPEECH RECOGNITION

**Experimental Setting.** We conduct automatic speech recognition (ASR) experiments on the LibriSpeech [1] dataset (Panayotov et al., 2015), which consists of audio recordings paired with their transcriptions. For implementation, we follow the recipes of SpeechBrain (Ravanelli et al., 2021) and describe the detailed settings in Appendix I.1. For our experiments, we use Branchformer (Peng et al., 2022), one of the state-of-the-art models for speech recognition, as well as a pure Transformer.

**Results.** Table 4 compares word error rates (WERs) on LibriSpeech 100h and 960h. For LibriSpeech 100h, Transformer+GFSA achieves 10.30/25.30 on the test clean/other set, which is a 6.53% improvement over the Transformer model for the WER of test clean. For LibriSpeech 960h, Transformer+GFSA shows a WER result of 2.31 in test clean, a 4.55% improvement over Transformer. Figure 7 in Appendix I.2 depicts the learning curves of train loss and valid loss when using GFSA, showing the effectiveness of our proposed filter.

## 4.4 EXPERIMENTS ON GRAPH TRANSFORMERS

**Experimental Setting.** To corroborate the efficacy of GFSA on graph transformers, we conduct experiments on a commonly used graph-level prediction dataset (i.e., ZINC) from benchmarking-

---

[1] http://www.openslr.org/12

Table 3: Compared with state-of-the-art models on ImageNet-1k dataset. The number in (↑) indicates the performance improvement over the base model.

| Category | Method | Input Size | #Layers | #Params | Top-1 Acc |
|---|---|---|---|---|---|
| CNN | ResNet-152 (He et al., 2016) | 224 | 152 | 230M | 78.1 |
| | DenseNet-201 (Huang et al., 2017) | 224 | 201 | 77M | 77.6 |
| CNN + Transformer | CVT-21 (Wu et al., 2021) | 224 | 21 | 32M | 82.5 |
| | Refiner (Zhou et al., 2021b) | 224 | 16 | 86M | 81.2 |
| Transformer | ViT-S/16 (Dosovitskiy et al., 2021) | 224 | 12 | 49M | 78.1 |
| | ViT-B/16 (Dosovitskiy et al., 2021) | 224 | 12 | 86M | 79.8 |
| | DeiT-S (Touvron et al., 2021a) | 224 | 12 | 22M | 79.8 |
| | DeiT-S Distilled (Touvron et al., 2021a) | 224 | 12 | 22M | 81.2 |
| | DeiT-S + LayerScale (Touvron et al., 2021b) | 224 | 12 | 22M | 80.5 |
| | DeiT-S + LateInsertion (Touvron et al., 2021b) | 224 | 12 | 22M | 80.5 |
| | DeiT-S + ClassAttention (Touvron et al., 2021b) | 224 | 12 | 22M | 80.6 |
| | DeiT-S + AttnScale (Wang et al., 2022) | 224 | 12 | 22M | 80.7 |
| | DeiT-S + FeatScale (Wang et al., 2022) | 224 | 12 | 22M | 80.9 |
| | DeiT-S + HAT (Bai et al., 2022) | 224 | 12 | 22M | 80.9 |
| | DeiT-S + Diverse (Chen et al., 2022) | 224 | 12 | 22M | 80.6 |
| | DeiT-S + ContraNorm (Guo et al., 2023) | 224 | 12 | 22M | 80.4 |
| | Swin-S (Liu et al., 2021) | 224 | 12 | 50M | 82.9 |
| | T2T-ViT-24 (Yuan et al., 2021) | 224 | 24 | 64M | 82.3 |
| | DeepViT-24B (Zhou et al., 2021a) | 224 | 24 | 36M | 80.1 |
| | DeiT-S (Touvron et al., 2021a) | 224 | 24 | 43M | 80.5 |
| | CaiT-S (Touvron et al., 2021b) | 224 | 24 | 47M | 82.6 |
| | DeiT-S + DiversePatch (Gong et al., 2021) | 224 | 24 | 44M | 82.2 |
| | DeiT-S + LayerScale (Touvron et al., 2021b) | 224 | 12 | 22M | 82.4 |
| | DeiT-S + AttnScale (Wang et al., 2022) | 224 | 24 | 44M | 81.1 |
| | DeiT-S + FeatScale (Wang et al., 2022) | 224 | 24 | 44M | 81.3 |
| | DeiT-S + ContraNorm (Guo et al., 2023) | 224 | 24 | 43M | 80.7 |
| GFSA | DeiT-S + GFSA | 224 | 12 | 43M | **81.1** (↑ 1.3) |
| | DeiT-S + GFSA | 224 | 24 | 43M | **81.5** (↑ 1.0) |
| | CaiT-S + GFSA | 224 | 24 | 43M | **82.8** (↑ 0.2) |
| | Swin-S + GFSA | 224 | 12 | 50M | **83.0** (↑ 0.1) |

Table 4: Results for ASR training on LibriSpeech 100h and 960h with GFSA

| Method | #Params | LibriSpeech 100h | | LibriSpeech 960h | |
|---|---|---|---|---|---|
| | | test-clean WER | test-other WER | test-clean WER | test-other WER |
| Transformer | 71.5M | 11.02 | 25.42 | 2.42 | 5.50 |
| Transformer + GFSA | 71.5M | **10.30** | **24.30** | **2.31** | **5.49** |
| Branchformer (Peng et al., 2022) | 109.8M | 9.63 | 22.43 | 2.13 | 5.00 |
| Branchformer + GFSA | 109.8M | **9.60** | **22.25** | **2.11** | **4.94** |

GNN leaderboards (Dwivedi et al., 2020) and the recent OGB-LSC quantum chemistry regression (i.e., PCQM4M-LSC) challenge (Hu et al., 2021), which is currently the biggest graph-level prediction dataset and contains more than 3.8M graphs in total. We choose Graphormer (Ying et al., 2021) as a backbone architecture and use its framework [2] for implementation. Mean absolute errors (MAEs) are reported for both regression tasks. For detailed experimental setting, see Appendix H.1.

**Results.** Tables 5 and 6 show that Graphormer with GFSA consistently outperforms the vanilla model across all datasets. As can be seen from the train MAE values of Table 6, using GFSA allows the model to learn more effectively during the training process. Notably, plugging GFSA improves the performance by a large margin, e.g., 7.20% relative validate MAE decline in PCQM4M. Due to space constraints, results with standard deviation are included in table 14 and 15 in Appendix H.2.

---

[2] https://github.com/microsoft/Graphormer

Table 5: Experimental results and number of parameters on ZINC

| Method | #Params | MAE |
|---|---|---|
| Graphormer | 500K | 0.1240 |
| Graphormer + GFSA | 500K | **0.1189** |

Table 6: Experimental results and number of parameters on PCQM4M and PCQM4Mv2

| Method | #Params | PCQM4M | | PCQM4Mv2 | |
|---|---|---|---|---|---|
| | | Train | Validate | Train | Validate |
| Graphormer | 48.3M | 0.0535 | 0.1286 | 0.0250 | 0.0862 |
| Graphormer + GFSA | 48.3M | **0.0312** | **0.1193** | **0.0249** | **0.0860** |

## 4.5 EXPERIMENTS ON CODE CLASSIFICATION

**Experimental Setting.** We conduct a code defect detection task based on Devign dataset provided by (Zhou et al., 2019). We use RoBERTa (Liu et al., 2019), CodeBERT (Feng et al., 2020), PLBART (Ahmad et al., 2021), and CodeT5 (Wang et al., 2021b) as our backbone models. The detailed settings are in Appendix J.1.

**Results.** Table 7 shows the accuracy of all models; GFSA results better than the base models. The biggest improvement is 2.40% for RoBERTa. In the case of CodeT5-base, using GFSA shows an accuracy of 64.75, an improvement of 1.95% from 63.51. CodeT5-small+GFSA has only about 100 additional parameters compared to CodeT5-small with 60M

Table 7: Code defect accuracy for models with plugged in GFSA. The number in (↑) indicates the improvement rate (%) over the base model.

| Method | Accuracy |
|---|---|
| RoBERTa (Liu et al., 2019) | 62.88 |
| RoBERTa + GFSA | **64.39** (↑ 2.40%) |
| CodeBERT (Feng et al., 2020) | 64.31 |
| CodeBERT + GFSA | **64.49** (↑ 0.12%) |
| PLBART (Ahmad et al., 2021) | 62.63 |
| PLBART + GFSA | **62.96** (↑ 0.52%) |
| CodeT5-small (Wang et al., 2021b) | 63.25 |
| CodeT5-small + GFSA | **63.69** (↑ 0.70%) |
| CodeT5-base (Wang et al., 2021b) | 63.51 |
| CodeT5-base + GFSA | **64.75** (↑ 1.95%) |

parameters, and even more impressively, it surpasses the accuracy of CodeT5-base. In Appendix J.2, we include case studies for this task. We also report the results of code clone detection task in Appendix K. In Table 17, CodeT5-small+GFSA outperforms CodeT5-small with an F1 of 94.36.

## 4.6 RUNTIME OVERHEADS

Our GFSA does not significantly increase training time. We summarize the runtime of base models vs. base models with GFSA in Appendix L. On GLUE tasks in natural language understanding, BERT with GFSA improves average performance from 82.51% to 83.58% (see Table. 1) with an overhead of less than 3 minutes in total training time. Considering the performance improvements, we think the increases in training time are negligible.

## 5 FINAL REMARKS

Our proposed graph filter-based self-attention (GFSA) mechanism achieves high performance with improvements on a variety of tasks in computer vision, natural language processing, graph pattern classification, speech recognition, and code classification. GFSA is a simple yet effective method that enriches the self-attention in Transformers with more diverse frequency information. This enables GFSA to address the oversmoothing problem and learn better latent representations for downstream tasks. However, our GFSA does not bring significant overheads in those Transformers' empirical runtime complexity. One can use more complicated graph filters to enhance accuracy more, but our goal is to find a balance between accuracy enhancements and overheads in runtime complexity.

We believe that GFSA is a promising new direction for improving Transformers. It is simple to implement and can be used in conjunction with other techniques to further improve the performance of Transformers. We hope that our work will inspire more research on graph filter-based self-attention and its applications.

## ETHICAL STATEMENTS

In terms of the broader impact of this research on society, we do not see the very negative impacts that might be expected. However, this paper may have implications for the carbon footprint and accessibility of learning algorithms. The computations required for machine learning research are rapidly growing, resulting in a larger carbon footprint (Schwartz et al., 2020). Our study improves performance and increases runtime very slightly, but the runtime increase is not very significant. However, in future research, it will also be important to study and improve our GFSA by taking carbon footprints into account.

GFSA improves the performance of existing Transformer-based models, which can have many positive impacts on society through services that utilize natural language processing, computer vision, and speech recognition. However, it will also be important to improve GFSA by considering other dimensions of AI, such as robustness to adversarial examples, fairness, and explainability.

## REPRODUCIBILITY STATEMENT

To ensure the reproducibility and completeness of this paper, we include the Appendix with 12 sections. Appendix A provides the complete proof of Theorem. 3.1. Appendix B contains the empirical study on the error related to Theorem. 3.1. Appendix C provides our PyTorch style pseudo code for our GFSA method. The pseudo code helps to implement our GFSA to any Transformers used a pure self-attention. All experiments in the paper are reproducible with additional implementation details provided in Appendices E to K. In an effort to ensure reproducibility, our GFSA code to reproduce the experiment can be found at `https://sites.google.com/view/gfsa-iclr/`.

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

# Appendix

## Table of Contents

# A PROOF OF THEOREM. 3.1

*Proof.* We begin by observing that the $L_\infty$ norm of $\bar{A}^2$ is bounded by:

$$\|\bar{A}^2\| \le \|\bar{A}\|^2. \tag{9}$$

By using the induction principle, it follows that for any positive integer $k$,

$$\|\bar{A}^k\| \le \|\bar{A}\|^k. \tag{10}$$

Given that $\bar{A}$ is a matrix normalized with softmax: i) all the elements of $\bar{A}$ lie within [0, 1], and ii) the sum of all the elements in $\bar{A}$ is equal to 1 (potentially less than 1 if an attention mask is used). Then we can calculate the upper bound of $L_\infty$ norm for $\bar{A}$ as follows:

$$\|\bar{A}\| = \max_i \sum_j |\bar{A}_{i,j}| \le \max_i \mathbf{1} = 1. \tag{11}$$

Now, considering the error term $E_K$ as given by Theorem 3.1, and applying the triangle inequality for matrix norms:

$$E_K = \|\bar{A}^K - (\bar{A} + (K-1)(\bar{A}^2 - \bar{A}))\| \tag{12}$$

$$\le \|\bar{A}^K\| + \|\bar{A}\| + (K-1)(\|\bar{A}^2\| + \|\bar{A}\|). \tag{13}$$

From the prior expressions, we deduce that $\|\bar{A}^K\| \le \|\bar{A}\|^K \le 1^K = 1$. Thus, the upper bound of the error term $E_K$ becomes as follows:

$$E_K \le 1 + 1 + (K-1)(1+1) = 2K. \tag{14}$$

$\square$

# B EMPIRICAL ERROR BOUND

In this section, we empirically compare the theoretical error bounds in Theorem. 3.1 with the actual errors. Figure 3 shows the theoretical upper bound of $E_K$ and the actual $E_K$ for the attention of the last layer in a trained pure backbone. DeiT-S and Graphormer are trained on ImageNet-1k and ZINC, respectively, and BERT is finetuned on STS-B. It can be observed that for all models, the actual $E_K$ is always smaller than the theoretical error bound.

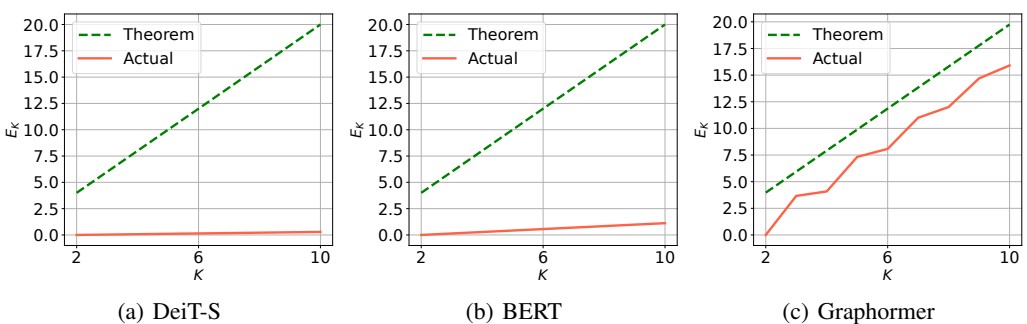

    (a) DeiT-S           (b) BERT          (c) Graphormer

Figure 3: Theoretical and actual values of $E_K$

## C    IMPLEMENTATION OF GFSA

The pseudo code of our GFSA is shown in Algorithm 1. For implememtation, $w_0$ and $w_1$ can be set as hyperparameters optionally.

---

**Algorithm 1** PyTorch-style pseudocode for GFSA

```
w_0 = torch.zeros(h)
w_1 = torch.ones(h)
w_K = torch.zeros(h)
I = torch.eyes(n)[None, None, ...]

def GFSA (att, K)
  att:  original self-attention
  att_K: high order term
  att_K = att + (K-1) * (torch.mm(att,att)-att)
  gf_att:  GFSA attention
  gf_att = w_0[None, :, None, None] * I
         + w_1[None, :, None, None] * att
         + w_K[None, :, None, None] * att_K
return gf_att
```

---

## D    OVERSMOOTHING AND ADDITIONAL VISUALIZATIONS

In Figure 2, we show the visualizations of oversmoothing characteristics in DeiT. We also provide visualizations in other domains. We show the filter response, cosine similarity, and singular value of BERT finetuned on STS-B dataset of GLUE tasks in Figure 4 and Graphormer finetuned on ZINC dataset in Figure 5.

To characterize self-attention, we first analyze the filter response of self-attention in the frequency domain. We follow the method used by Wang et al. (2022) for spectral visualization of the self-attention matrix. As shown in Figure 2 (a), Deit has a near-zero magnitude for the high frequencies, which is characteristic of a low-frequency filter and is likely to result in oversmoothing when applied multiple times.

We follow the calculation method of Guo et al. (2023) for cosine similarity. As shown in Figure 2 (b), the higher similarity as the layers of the model get deeper is related to the oversmoothing problem. To further analyze this issue, we also consider the dimensionality collapse in Transformer-based models. We plot the singular value distribution of the feature in the last block. As shown in Figure 2 (c), insignificant, near-zero values dominate the feature distribution. As layers get deeper, the similarity of features increases and dimensional collapse occurs. The oversmoothing problem is the same in BERT and Graphormer, as shown in Figure 4 and Figure 5.

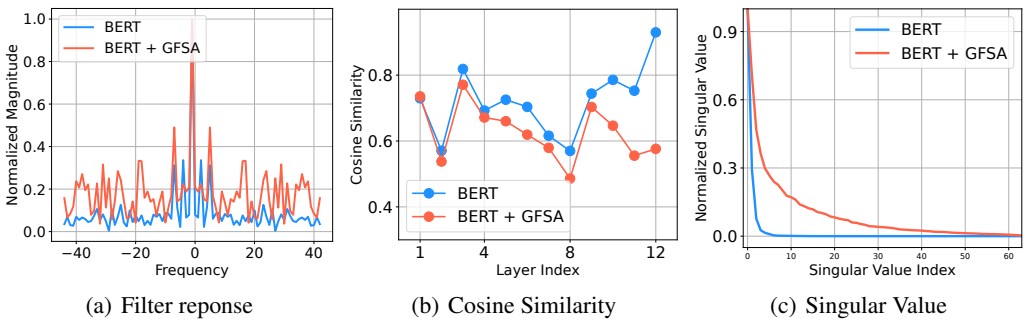

(a) Filter reponse          (b) Cosine Similarity          (c) Singular Value

Figure 4: Filter frequency response, cosine similarity, and singular values on STS-B for BERT and BERT+GFSA

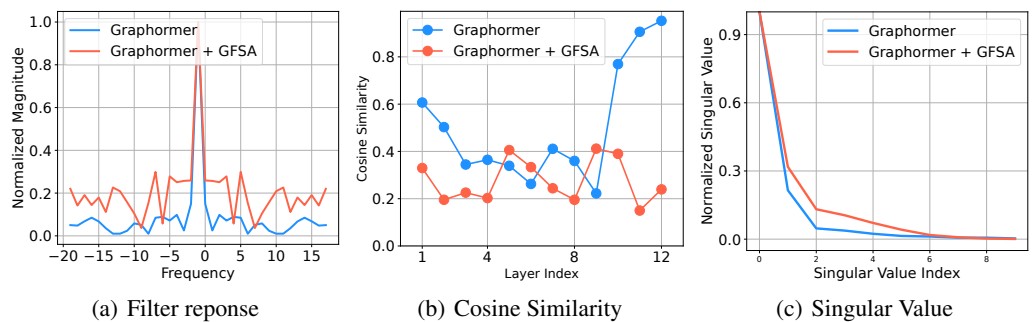

(a) Filter reponse  (b) Cosine Similarity  (c) Singular Value

Figure 5: Filter frequency response, cosine similarity, and singular values on ZINC for Graphormer and Graphormer+GFSA

# E    NATURAL LANGUAGE UNDERSTANDING

## E.1    DETAILED EXPERIMENTAL SETTINGS

**Dataset.**    The benchmark datasets we used are listed below.

CoLA    The Corpus of Linguistic Acceptability (Warstadt et al., 2019) consists of English acceptability judgments drawn from books and journal articles. The target task is a binary classification task, and each sentence is determined to be grammatically acceptable or not.

SST-2    The Stanford Sentiment Treebank (Socher et al., 2013) is a dataset in which each sentence is sourced from movie reviews and accompanied by human annotations of their sentiment. The target task is binary sentiment classification for a single sentence.

MRPC    The Microsoft Research Paraphrase Corpus (Dolan & Brockett, 2005) is a corpus of sentence pairs, which are automatically extracted from online news sources and annotated by humans. The target is to determine whether the sentences in the pair are semantically equivalent.

QQP    The Quora Question Pairs (Chen et al., 2018) dataset is a collection of question pairs from the community question-answering website Quora. The target is to determine whether the questions in the pair are semantically equivalent.

STS-B    The Semantic Textual Similarity Benchmark (Cer et al., 2017) is a collection of sentence pairs drawn from news headlines, video and image captions, and natural language inference data with human annotation. The target is a regression task to predict a similarity score from 0 to 5.

MNLI    The Multi-Genre Natural Language Inference Corpus (Williams et al., 2017) is a crowd-sourced collection of sentence pairs with textual entailment annotations. Given a premise sentence and a hypothesis sentence, the task is to predict whether the premise entails the hypothesis (entailment), contradicts the hypothesis (contradiction), or neither (neutral). The standard test set consists of private labels from the authors and evaluates both the matched (in-domain) and mismatched (cross-domain) sections.

QNLI    The Stanford Question Answering (Wang et al., 2018) dataset is a question-answering dataset consisting of question-paragraph pairs, where one of the sentences in the paragraph contains the answer to the corresponding question written by an annotator. The task is to determine whether the context sentence contains the answer to the question.

RTE    The Recognizing Textual Entailment (Bentivogli et al., 2009) dataset comes from a series of annual textual entailment challenges. The target task is a binary entailment classification task.

**BERT.**    BERT (Devlin et al., 2019) consists with 12 layers, 12 heads, 768 hidden size, 512 maximum sequence length, and MLP dimension of 3072.

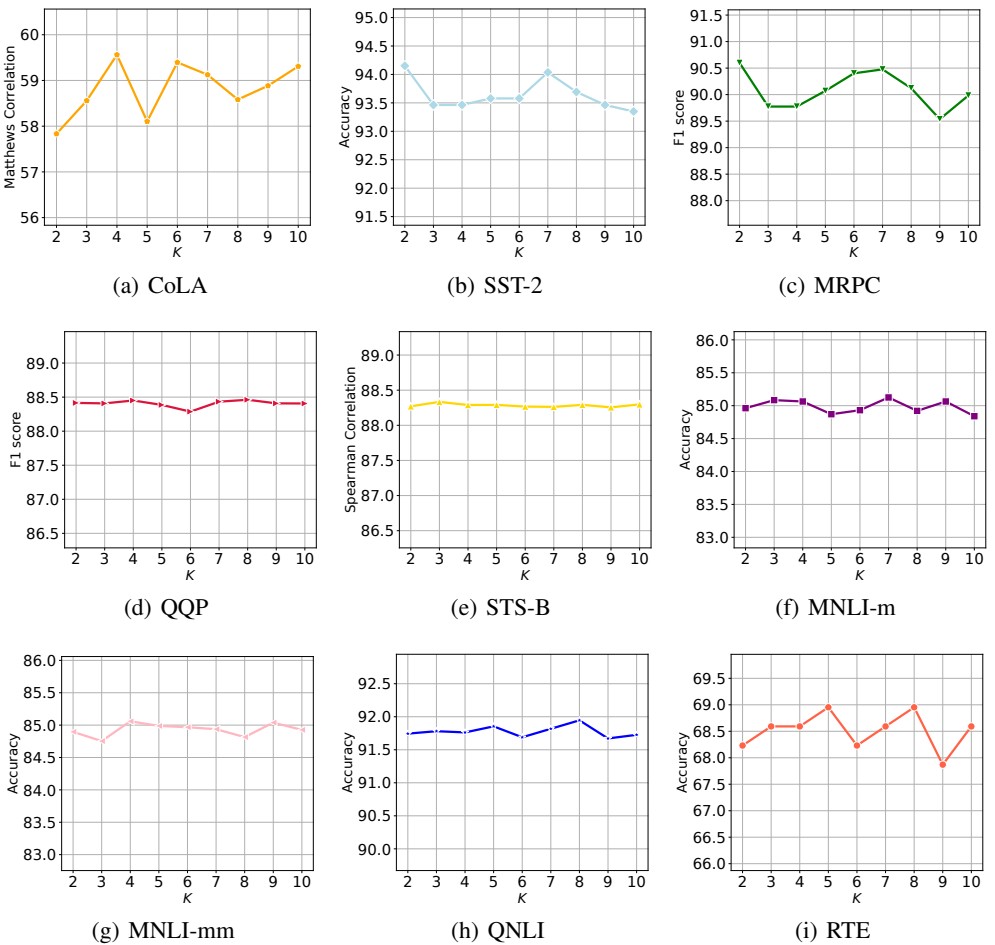

Figure 6: Sensitivity results on various $K$ with BERT$_{\text{BASE}}$ finetuned on GLUE tasks

**ALBERT.** ALBERT (Lan et al., 2019) consists with 12 layers, 12 heads, 768 hidden size, 512 maximum sequence length, 128 embedding dimension, and MLP dimension of 3072.

**RoBERTa.** RoBERTa (Liu et al., 2019) consists with 12 layers, 12 heads, 768 hidden size, 514 maximum sequence length, and MLP dimension of 3072.

**Training.** For implementation, we adopt HuggingFace framework. We trained all models with 5 epochs with 32 batch size. The linear learning rate decay is used and initial learning rate is set to $2 \times 10^{-5}$. We use AdamW (Loshchilov & Hutter, 2017) optimizer, and weight decay is set to 0. All models are trained on 1 GPU and of NVIDIA RTX A5000 24GB.

### E.2 SENSITIVITY TO $K$

In this section, we explore the influence of the polynomial order, denoted as $K$, in our GFSA, conducting experiments on BERT$_{\text{BASE}}$ finetuned with GLUE tasks. We search for values of $K$ from 2 to 10, and the results are presented in Figure 6. For each dataset, Optimal $K$ exists and the performance of models using GFSA is generally robust to changes in $K$.

# F  CAUSAL LANGUAGE MODELING

## F.1  DETAILED EXPERIMENTAL SETTINGS

**Dataset.**  The benchmark datasets we used are listed below.

PTB  Penn Treebank (Marcus et al., 1993) dataset is a collection of text documents that have been extensively annotated with linguistic information, primarily syntactic and grammatical structures.

WIKITEXT  WikiText (Merity et al., 2016) dataset is a collection of over 100 million tokens extracted from the set of verified good and featured articles on Wikipedia. Compared to the preprocessed version of Penn Treebank (PTB), WikiText-2 is over 2 times larger and WikiText-103 is over 110 times larger.

**GPT2.**  GPT-2 (Radford et al., 2019) is a Transformer pretrained on a very large corpus of English data in a self-supervised fashion without any human labelling on dataset. It automatically generate inputs and labels from those texts, and trained to guess the next word in sentences. For implementation, we adopt HuggingFace Framework [3]. For all experiments, GPT2 has 12 layers with 12 attention heads, 768 hidden size and 1024 maximum sequence length, resulting in a total of 117 million parameters.

**Training.**  We finetune GPT2 with 4 batch size, $5 \times 10^{-5}$ learning rate and linear learning weight decay using adamW (Loshchilov & Hutter, 2017) optimizer. We also apply dropout with probability 0.1. Following (Yao et al., 2022), we train models for 15 epochs with PTB, 4 epochs with WikiText-103 and 10 epochs with WikiText-2. We use sensitivity metric, i.e., perplexity, which is a commonly used metric to evaluate the performance of language models, particularly in language modeling and text generation tasks. perplexity measures how well a language modeling can predict a sequence of words in a given text or a test dataset. All the experiments are conducted on 1 GPU and of NVIDIA RTX 3090 24GB.

## F.2  SENSITIVITY TO $K$

We conducted a sensitivity study on $K$ of GPT-2 across all datasets, and the results are presented in Table 8. For PTB and WikiText-2, GFSA exhibits the best performance when $K$ is high, typically around 8 or 9. However, for WikiText-103, GFSA achieves the best perplexity when $K$ is small, specifically when $K$ is 3 or 4.

Table 8: Results comparison on GPT-2 finetuned with GFSA

| Method | #Params | PTB | WikiText-2 | WikiText-103 |
|---|---|---|---|---|
| GPT2 (Radford et al., 2019) | 117M | 19.513 | 20.966 | 15.939 |
| GPT2 + GFSA($K = 2$) | 117M | 19.459 | 20.929 | 15.920 |
| GPT2 + GFSA($K = 3$) | 117M | 19.456 | 20.927 | **15.919** |
| GPT2 + GFSA($K = 4$) | 117M | 19.453 | 20.927 | **15.919** |
| GPT2 + GFSA($K = 5$) | 117M | 19.452 | 20.925 | 15.920 |
| GPT2 + GFSA($K = 6$) | 117M | 19.451 | 20.925 | 15.920 |
| GPT2 + GFSA($K = 7$) | 117M | **19.450** | 20.925 | 15.921 |
| GPT2 + GFSA($K = 8$) | 117M | **19.450** | 20.924 | 15.921 |
| GPT2 + GFSA($K = 9$) | 117M | **19.450** | **20.923** | 15.921 |

---

[3] https://github.com/huggingface/transformers

## G  IMAGE CLASSIFICATION

### G.1  DETAILED EXPERIMENTAL SETTINGS

Our code is implemented based on the timm library (Wightman, 2019). In the case of our training recipe, it is the same as experimental setting of Wang et al. (2022) that follows the training recipes of Touvron et al. (2021a) and Touvron et al. (2021b). To apply our GFSA to existing base models such as DeiT, Cait, and Swin, we consider a range of $K$ between 2 and 5. For 12-layer DeiT, we follow the same hyperparameters from Wang et al. (2022). We set the dropout rate to 0 and 0.2 for 12-layer and 24-layer DeiT, respectively. For CaiT, we apply our GFSA on only to the patch embedding layer. All other hyper-parameters are kept consistent with the original papers of DeiT Touvron et al. (2021a), CaiT (Touvron et al., 2021b) and, Swin (Liu et al., 2021). All models are trained on NVIDIA RTX 3090 24GB.

### G.2  FLOPS & THROUGHPUT

In Table 9, we report the number of FLOPs and throughput. With GFSA plugged in, the FLOP count is either the same or no different. For DeiT-S with 24 layers, which shows a slight FLOP increase with GFSA plugged in. However, for the rest of the settings, the models have the same number of Flops. For throughput, it tends to decrease because calculating the high-order term is an additional cost.

Table 9: Experimental evalutation of GFSA plugged into DeiT-S, CaiT-S, and Swin-S

| Backbone | Method | Input Size | #Layers | #Params | #FLOPs | #Throughput | Top-1 Acc |
|---|---|---|---|---|---|---|---|
| DeiT | DeiT-S | 224 | 12 | 22.0M | 4.57G | 856.07 | 79.8 |
| | DeiT-S + GFSA | 224 | 12 | 22.0M | 4.57G | 614.54 | **81.1** (↑ 1.3) |
| | DeiT-S | 224 | 24 | 43.3M | 9.09G | 423.68 | 80.5 |
| | DeiT-S + GFSA | 224 | 24 | 43.3M | 9.10G | 314.75 | **81.5** (↑ 1.0) |
| CaiT | CaiT-S | 224 | 24 | 46.9M | 9.34G | 574.66 | 82.6 |
| | CaiT-S + GFSA | 224 | 24 | 47.0M | 9.34G | 406.96 | **82.8** (↑ 0.2) |
| Swin | Swin-S | 224 | 24 | 49.6M | 8.75G | 912.38 | 82.9 |
| | Swin-S + GFSA | 224 | 24 | 49.6M | 8.75G | 714.60 | **83.0** (↑ 0.1) |

### G.3  SENSITIVITY TO $K$

We also perform the sensitivity analysis for $K$. Tables 10 and 11 show the results of sensitivity analysis for DeiT-S and CaiT-S with GFSA plugged in. For 12-layer DeiT-S, GFSA performance of 81.12 is highest when $K = 3$. When the GFSA has a $K$ of 2, the performance is worse than the original DeiT-S, but when the $K$ is 3 or higher, the performance is better than the original DeiT-S, and most surprisingly, the performance is better than the 24-layer DeiT-S.

CaiT-S shows the highest performance of 82.84 when $K = 4$. For CaiT-S, the accuracy is slightly lower than that of the original CaiT-S when $K = 2$, but it starts to exceed the accuracy of CaiT-S when $K$ is 3 or higher.

Table 10: Sensitivity to $K$ for 12-layer DeiT-S + GFSA

| $K$ | 2 | 3 | 4 | 5 |
|---|---|---|---|---|
| Top-1 Acc (%) | 79.27 | **81.12** | 80.86 | 81.07 |

Table 11: Sensitivity to $K$ for 24-layer CaiT-S + GFSA

| $K$ | 2 | 3 | 4 |
|---|---|---|---|
| Top-1 Acc (%) | 82.54 | 82.65 | **82.84** |

## G.4 ADDITIONAL EXPERIMENTS WITH GUO ET AL. (2023)'S SETTING

To make a fair comparison with ContraNorm (Guo et al., 2023), one of the related studies that mitigates oversmoothing, we run additional experiments to match their experimental setup.

**Experimental Setting.** We follow the training recipe used by Guo et al. (2023), which is a slightly modified version of Touvron et al. (2021a)'s recipe. Guo et al. (2023) use AdamW optimizer with cosine learning rate decay. We select the DeiT-T and DeiT-S for ImageNet-100 and ImageNet-1k, respectively. "T" and "S" denote tiny and small model sizes, respectively. For all experiments, the image size is set to be 224x224. We train each model for 300 epochs and the batch size is set to 1024. For ContraNorm, we train with their recommended hyperparameters. All models are trained on 4 GPUs and of NVIDIA RTX A6000 48GB.

**Experimental Results.** In Table 12, DeiT-T and DeiT-S with GFSA outperform vanilla DeiT-T and DeiT-S in all layer settings. On ImageNet-100, GFSA improves the performance of DeiT-T with 12 layers by 1.52%. The largest gain is a 4.88% improvement on 16-layer DeiT-T. This shows that the effect of GFSA is larger than the effect of ContraNorm. For ImageNet-1k, surprisingly, with 16 layers, GFSA is able to increase the performance of DeiT-S by 80.83%, meaning that GFSA brings benefits with a 3.23% improvement.

Table 12: Experiment results on ImageNet-100 and ImageNet-1k

| Dataset | Method | #Layers=12 | #Layers=16 | #Layers=24 |
|---|---|---|---|---|
| ImageNet-100 | DeiT-T | 76.52 | 75.34 | 76.76 |
| | DeiT-T + ContraNorm | 77.03 | 78.72 | 78.12 |
| | DeiT-T + GFSA | **77.68** | **79.02** | **78.64** |
| ImageNet-1k | DeiT-S | 77.32 | 78.25 | 77.69 |
| | DeiT-S + ContraNorm | 77.80 | 79.04 | 78.67 |
| | DeiT-S + GFSA | **79.86** | **80.83** | **79.15** |

**Sensitivity to $K$.** In Table 13, we experiment with a sensitivity analysis for $K$. For ImageNet-100, the performance of GFSA generally improves when $K$ is 4 or 5. On the other hand, GFSA performs better at lower $K$ for settings that are layers 16 and 24 for ImagNet-1k.

Table 13: Experiment results on ImageNet-100 and ImageNet-1k

| Dataset | Method | $K$ | #Layers=12 | #Layers=16 | #Layers=24 |
|---|---|---|---|---|---|
| ImageNet-100 | DeiT-T + GFSA | 2 | 76.92 | 78.14 | 78.40 |
| | DeiT-T + GFSA | 3 | 77.41 | 77.76 | 78.09 |
| | DeiT-T + GFSA | 4 | 77.01 | **79.02** | **78.64** |
| | DeiT-T + GFSA | 5 | **77.68** | 78.14 | **78.64** |
| ImageNet-1k | DeiT-S + GFSA | 2 | 79.84 | **80.83** | **79.15** |
| | DeiT-S + GFSA | 3 | 79.85 | 79.39 | 79.07 |
| | DeiT-S + GFSA | 4 | **79.86** | 79.44 | 79.10 |

# H    GRAPH CLASSIFICATION

## H.1    DETAILED EXPERIMENTAL SETTINGS

**Dataset.**    The benchmark datasets we used are listed below.

ZINC    ZINC (Irwin et al., 2012) is the most popular real-world molecular dataset to predict graph property regression for constrained solubility, an important chemical property for designing generative GNNs for molecules. Uniform sampling is adopted for data splitting. We use a ZINC-subset of small-scale dataset.

PCQM4M    PCQM4M-LSC  (Hu et al., 2021) is 2D molecular graphs, which is one of the most practically relevant quantum chemical properties of molecule science. The task is to predict DFT(density functional theory)-calculated HOMO-LUMO energy gap of molecules given their graphs. PCQM4M-LSC is unprecedentedly large in scale comparing to other labeled graph-level prediction datasets, which contain more than 3.8M graphs. We use PCQM4M and PCQM4Mv2 large-scale datasets.

**Graphormer.**    Following Graphormer (Ying et al., 2021), we use Graphormer for PCQM4M and Graphormer$_{\text{SLIM}}$ for ZINC. Graphormer consists of 12 encoder layers, 80 encoder embedding dimension, and 768 MLP dimension. It employs 32 encoder heads and 24 hidden dimension for each head. Graphormer$_{\text{SLIM}}$ consists of 12 encoder layers, 80 encoder embedding dimension, and 80 MLP dimension. It employs 8 encoder heads and 10 hidden dimension for each head.

**Training.**    We adopt a training recipe from Graphormer (Ying et al., 2021).   We use adamW (Loshchilov & Hutter, 2017) optimizer with 0.9 and 0.999 coefficients for running averages of gradient and its square, and use Mean Absolute Error (MAE) as loss function. We use polynomial learning rate decay, with initial learning rate set to $2 \times 10^{-4}$ and end learning rate set to $1 \times 10^{-9}$. For ZINC, we set batch size as 256, max epochs as 10k, and warm-up stage step as 40k. For PCQM4M and PCQM4Mv2, we set batch size as 1024, max epochs as 300, and warm-up stage step as 60k. All models are trained on 1 GPU and of NVIDIA RTX 3090 24GB. We conduct experiments with 4 different seeds.

## H.2    EXPERIMENTAL RESULTS WITH STANDARD DEVIATION

Table 14: Experimental results and number of parameters on ZINC

| Method | #Params | MAE |
|---|---|---|
| Graphormer | 500K | 0.1240±0.006 |
| Graphormer + GFSA | 500K | **0.1189±0.002** |

Table 15: Experimental results and number of parameters on PCQM4M and PCQM4Mv2

| Method | #Params | PCQM4M | | PCQM4Mv2 | |
|---|---|---|---|---|---|
| | | Train | Validate | Train | Validate |
| Graphormer | 48.3M | 0.0535±0.038 | 0.1286±0.016 | 0.0250±0.000 | 0.0862±0.000 |
| Graphormer + GFSA | 48.3M | **0.0312±0.001** | **0.1193±0.000** | **0.0249±0.000** | **0.0860±0.000** |

We conducted experiments following the experimental environments of Graphormer Ying et al. (2021) using 4 different seeds. Due to space constraints, only the mean values are reported in Tables 5 and 6. In this section, we report the results with mean and standard deviations in Tables 14 and 15.

# I  AUTOMATIC SPEECH RECOGNITION

## I.1  DETAILED EXPERIMENTAL SETTINGS

**Dataset.**  We conduct experiments on the LibriSpeech [4] dataset (Panayotov et al., 2015), which consists of audio recordings paired with their transcriptions. The LibriSpeech dataset has approximately 1,000 hours of read English speech with a sampling rate of 16 kHz. We keep the original 16,000Hz sampling rate and compute 80-dim log-Mel filterbanks for a 25ms sliding window, strided by 10ms. The filterbank features are then normalized to zero mean and unit variance per input sequence. For implementation, we follow the recipes of SpeechBrain (Ravanelli et al., 2021).

**Evaluation Metric.**  Word error rate (WER (%)) is derived from the Levenshtein distance and compares a reference to a hypothesized word-level transcription. It is calculated by summing the number of word insertions, deletions, substitutions and dividing it by the total number of words in the reference transcription.

**Vanilla Transformer.**  We use a vanilla Transformer to apply our GFSA. For implementation, we use a SpeechBrain (Ravanelli et al., 2021) framework. The vanilla Transformer consists of i) 1D convolution to perform striding, ii) Transformer encoder with 12 layers, 4 heads, embedding dimension of 512, MLP dimension of 2048, and post-LayerNorm iii) decoder with 6 layers, 4 heads, embedding dimension of 512, MLP dimension of 2048, joint beamsearch, and iv) external Transformer language model with 12 layers, 12 heads, embedding dimension of 768, and MLP dimension of 3072.

**Branchformer.**  We use one of the SOTA models, Branchformer (Peng et al., 2022) to plug-in our GFSA. Branchformer has two parallel branches, one for capturing global interactions using attention and the other for more localized context using convolutional gating MLP. The Branchformer architecture for speech recognition consists of i) 1D convolution to perform striding, ii) Branchformer encoder with 18 layers, 8 heads, embedding dimension of 512, and MLP dimension of 3072, iii) decoder with 6 layers, 8 heads, embedding dimension of 512, a convolutional spatial gating unit (CSGU) dimension of 3072, joint beamsearch, and iv) external Transformer language model with 12 layers, 12 heads, embedding dimension of 768, and MLP dimension of 3072.

**Training.**  We follow a training recipe from SpeechBrain (Ravanelli et al., 2021). The standard LibriSpeech validation sets (dev-clean and dev-other) are used to tune all parameters and select the best models. Test sets (test-clean and test-other) are used only to report final WER performance. We train the pure Transformer for 100 epochs and the Branchformer for 120 epochs with a batch size of 16. We use a data augmentation method on all models using SpecAugment (Park et al., 2019). SpecAugment applies time and frequency masking as well as time warping to the input spectrum. For Branchformer, we use AdamW (Loshchilov & Hutter, 2017) optimizer with 0.9 and 0.98 coefficients for computing running averages of gradient and its square. The learning rate and weight decay in all models are 0.0008 and 0.01, respectively. We use a connectionist temporal classification (CTC) loss Graves & Jaitly (2014); Zhang et al. (2016). We also apply dropout with probability 0.1 and label smoothing with weight 0.1 to mitigate overfitting. We fix the random seed as 74443 on all experiments. All models are trained on 1 GPU and of NVIDIA RTX A6000 48GB.

**Hyperparameters.**  In Table 16, we describe main hyperparameters used in the automatic speech recognition task. For Transformer+GFSA and Branchformer+GFSA, we also report the best $K$ hyperparameter.

---

[4] http://www.openslr.org/12

Table 16: Main hyperparameters used in ASR

| Model | Experimental Setting |
|---|---|
| Transformer | Encoder: Transformer (12 layers)
Decoder: Transformer (6 layers) + (CTC/ATT joint) beamsearch + TransformerLM
Augmentation: SpecAugment
Features: 40 fbanks
Pretraining: no
Dropout: 0.1
Batchnorm: yes
Number of epochs: 100
Batch size: 32
Learning rate: 0.0008
LR scheduler: Noam
Optimizer: Adam
Loss: CTC + KLdiv (Label Smoothing loss)
CTC weight: 0.3 |
| Transformer+GFSA | Encoder: Transformer (12 layers)
Decoder: Transformer (6 layers) + (CTC/ATT joint) beamsearch + TransformerLM
Augmentation: SpecAugment
Features: 40 fbanks
Pretraining: no
Dropout: 0.1
Batchnorm: yes
Number of epochs: 100
Batch size: 32
Learning rate: 0.0008
LR scheduler: Noam
Optimizer: Adam
Loss: CTC + KLdiv (Label Smoothing loss)
CTC weight: 0.3
$K$: 2 |
| Branchformer | Encoder: Branchformer
Decoder: Transformer (6 layers) + (CTC/ATT joint) beamsearch + TransformerLM
Augmentation: SpecAugment
Features: 40 fbanks
Pretraining: no
Dropout: 0.1
Batchnorm: yes
Number of epochs: 120
Batch size: 16
Learning rate: 0.0008
LR scheduler: Noam
Optimizer: AdamW with coefficients 0.9 and 0.98
Loss: CTC + KLdiv (Label Smoothing loss)
CTC weight: 0.3 |
| Branchformer+GFSA | Encoder: Branchformer
Decoder: Transformer (6 layers) + (CTC/ATT joint) beamsearch + TransformerLM
Augmentation: SpecAugment
Features: 40 fbanks
Pretraining: no
Dropout: 0.1
Batchnorm: yes
Number of epochs: 120
Batch size: 16
Learning rate: 0.0008
LR scheduler: Noam
Optimizer: AdamW with coefficients 0.9 and 0.98
Loss: CTC + KLdiv (Label Smoothing loss)
CTC weight: 0.3
$K$: 3 |

## I.2 Training Curve

We compare the training and validation curves for LibriSpeech 100h in Figure 7. The training loss curve of GFSA is lower than the pure Transformer. GFSA stabilizes the loss curve of pure Transformer slightly earlier.

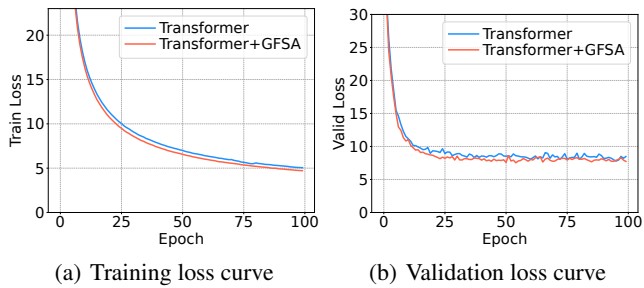

(a) Training loss curve        (b) Validation loss curve

Figure 7: Training curve on LibriSpeech 100h

## J Code Defect Detection

### J.1 Detailed Experimental Settings

**Dataset.** We use Devign dataset provided by (Zhou et al., 2019), which is a binary classification task to evaluate whether a C language code is vulnerable to software systems or not.

**Implementation.** We build our experiments on top of the open-sourced code [5] and recipes provided by Wang et al. (2021b).

**RoBERTa.** RoBERTa (Liu et al., 2019) is an encoder-only model trained with masked language modeling on code. All hyperparameters are consistent with the training method in the source code of Wang et al. (2021b).

**PLBART.** PLBART (Ahmad et al., 2021) is an encoder-decoder model based on BART (Lewis et al., 2020) architecture. PLBART can support understanding and generation tasks. All hyperparameters are consistent with the training method in the source code of Wang et al. (2021b).

**CodeBert.** CodeBERT (Feng et al., 2020) is a model trained on masked language modeling and replaced token detection. CodeBERT is a bimodal pretrained model based on Transformer with 12 layers for programming language and natural language. All hyperparameters are consistent with the training method in the source code of Wang et al. (2021b).

**CodeT5.** CodeT5 is an encoder-decoder framework with the same architecture as T5 (Raffel et al., 2020). It aims to derive generic representations for programming language and natural language via pre-training on unlabeled source code. CodeT5-small has 6 encoder layers, 6 decoder layers, 8 attention heads, 512 dimensional hidden states, and 60M parameters. The other models have 12 encoder layers, 12 decoder layers, 12 attention heads, 768 dimensional hidden states, and 220M parameters. All hyperparameters are consistent with the training method in the source code of Wang et al. (2021b).

**Training.** The pre-trained models mentioned above are applied to this downstream task. We add GFSA directly on top of self-attention. We finetune baselines and GFSA models for 10 epochs with a batch size of 16. We use early stopping strategy with a patience of 2. Models generate binary labels from unigram sequences at the decoder for defect detection task. We employ accuracy for evaluating the code defect detection task. All models are trained on 1 GPU and of NVIDIA RTX A6000 48GB.

---
[5] https://github.com/salesforce/CodeT5

## J.2 CASE STUDY

In Listing 1, we show one of case for code snippets of defects in QEMU [6] that CodeT5-base does not predict correctly, but that *only* CodeT5-base+GFSA predicts. The commit message [7] for this case is as follow:

> Needed for changing `cpu_has_work()` argument type to CPUState, used in `h_cede()`.

`h_cede()` is the hypercall that asks the hypervisor to shut down the CPU. Previously, this hypercall simply passed the CPUID, so the hypervisor did not know what state the CPU was in. This change allows the hypervisor to know whether the CPU is actually performing work. If the CPU is performing a task, the hypervisor waits for the CPU to complete the task.

In this context, accurately predicting defects like the one above is very important, and applying GFSA to CodeT5-base helps in terms of performance improvement.

```
1  @@ -204,7 +204,7 @@ static target_ulong put_tce_emu(sPAPRTCETable *tcet,
       target_ulong ioba,
2  - static target_ulong h_put_tce(CPUPPCState *env, sPAPREnvironment *spapr
3  + static target_ulong h_put_tce(PowerPCCPU *cpu, sPAPREnvironment *spapr
4                                 , target_ulong opcode, target_ulong *args)
5   {
6    target_ulong liobn = args[0];
7    target_ulong ioba = args[1];
8    target_ulong tce = args[2];
9    VIOsPAPRDevice *dev = spapr_vio_find_by_reg(spapr->vio_bus, liobn);
10   VIOsPAPR_RTCE *rtce;
11   if (!dev) {
12     hcall_dprintf("LIOBN 0x" TARGET_FMT_lx " does not exist\n", liobn);
13     return H_PARAMETER;
14   }
15   ioba &= ~(SPAPR_VIO_TCE_PAGE_SIZE - 1);
16   #ifdef DEBUG_TCE
17     fprintf(stderr, "spapr_vio_put_tce on %s ioba 0x" TARGET_FMT_lx " TCE
       0x" TARGET_FMT_lx "\n", dev->qdev.id, ioba, tce);
18   #endif
19   if (ioba >= dev->rtce_window_size) {
20     hcall_dprintf("Out-of-bounds IOBA 0x" TARGET_FMT_lx "\n", ioba);
21     return H_PARAMETER;
22   }
23   rtce = dev->rtce_table + (ioba >> SPAPR_VIO_TCE_PAGE_SHIFT);
24   rtce->tce = tce;
25   return H_SUCCESS;
26  }
```

Listing 1: An example commit history for defects in Devign dataset

## K CODE CLONE DETECTION

### K.1 DETAILED EXPERIMENTAL SETTINGS

**Dataset.** Code clone detection aims to measure the similarity between two code snippets and predict whether they have the same functionality. We experiment with the Java data provided by Wang et al. (2020).

**Implementation.** We build our experiments on top of the open-sourced code [8] and recipes provided by Wang et al. (2021b).

---

[6] https://www.qemu.org
[7] https://github.com/qemu/qemu/commit/b13ce26d3e8c6682044ae84920f2417b30ce356b
[8] https://github.com/salesforce/CodeT5

**Training.** We finetune both CodeT5 and CodeT5+GFSA for one epoch with a batch size of 16. We also use early stopping with patience of 2. CodeT5 and CodeT5+GFSA encode source code and take the representation to calculate similarity of two code snippets. We employ F1 score for evaluating this task. All models are trained on 1 GPU and of NVIDIA RTX A6000 48GB.

## K.2 EXPERIMENT RESULT

Table 17 shows results comparing CodeT5 and CodeT5 with GFSA. The result shows that by using our GFSA, CodeT5 models improve their performance. CodeT5-small+GFSA provides a 0.61% improvment over Code5T-small.

Table 17: Results on the code clone detection task

| Method | Clone F1 |
|---|---|
| CodeT5-small (Wang et al., 2021b) | 94.36 |
| CodeT5-small + GFSA | **94.94** (↑ 0.61%) |
| CodeT5-base (Wang et al., 2021b) | 94.31 |
| CodeT5-base + GFSA | **94.92** (↑ 0.64%) |

## L TIME COMPLEXITY AND EMPIRICAL RUNTIME ANALYSIS

**Time Complexity.** The time complexity of original self-attention is $\mathcal{O}(n^2 d)$. But our GFSA has a high order term. Therefore, the time complexity of GFSA has $\mathcal{O}(n^2 d + n^3)$. If $n$ is smaller than $d$, the time complexity approaches $\mathcal{O}(n^2 d)$, which is the complexity of original self-attention.

**Empirical Runtime Analysis.** We report the training time of various methods with GFSA in Tables 18 to 23. In general, the training time of methods with GFSA is slightly longer than that of existing methods. For example, the Transformer for the automatic speech recognition task increases from 190.5 seconds to 191.6 seconds on Librispeech 100h dataset, as increases of only 1 second. Instead of computing higher-order polynomial terms, our GFSA approximates them, with only a small increase in runtime, which is not very significant.

Table 18: Training time (seconds per epoch) on GLUE tasks. $s$ denotes the abbreviation for second. **Avg** denotes the average training time across all tasks.

| Datasets | #Params | CoLA | SST-2 | MRPC | QQP | STS-B | MNLI-m/mm | QNLI | RTE | **Avg** |
|---|---|---|---|---|---|---|---|---|---|---|
| BERT$_{\text{BASE}}$ (Devlin et al., 2019) | 110M | $17s$ | $182s$ | $17s$ | $1483s$ | $24s$ | $2004s$ | $580s$ | $18s$ | $541s$ |
| BERT$_{\text{BASE}}$ + GFSA | 110M | $19s$ | $192s$ | $19s$ | $1571s$ | $25s$ | $2147s$ | $621s$ | $20s$ | $577s$ |
| ALBERT$_{\text{BASE}}$ (Lan et al., 2019) | 11M | $15s$ | $188s$ | $20s$ | $1506s$ | $25s$ | $2072s$ | $612s$ | $19s$ | $557s$ |
| ALBERT$_{\text{BASE}}$ + GFSA | 11M | $16s$ | $197s$ | $21s$ | $1604s$ | $26s$ | $2219s$ | $659s$ | $21s$ | $595s$ |
| RoBERTa$_{\text{BASE}}$ (Liu et al., 2019) | 125M | $17s$ | $190s$ | $18s$ | $1492s$ | $25s$ | $2012s$ | $593s$ | $18s$ | $546s$ |
| RoBERTa$_{\text{BASE}}$ + GFSA | 125M | $19s$ | $200s$ | $19s$ | $1580s$ | $26s$ | $2151s$ | $634s$ | $20s$ | $581s$ |

Table 19: Training time (seconds per epoch) on causal language modeling tasks.

| Method | #Params | PTB | WikiText-2 | WikiText-103 | **Avg** |
|---|---|---|---|---|---|
| GPT2 (Radford et al., 2019) | 117M | $89.1s$ | $195.7s$ | $9638.4s$ | $3307.8s$ |
| GPT2 + GFSA | 117M | $160.3s$ | $354.2s$ | $17424.6s$ | $5979.7s$ |

Table 20: Training time (seconds per epoch) on ImageNet-1k

| Backbone | Method | #Layers | #Params | #FLOPs | #Throughput | Runtime |
|---|---|---|---|---|---|---|
| DeiT | DeiT-S | 12 | 22.0M | 4.57G | 856.07 | 1900.9$s$ |
| | DeiT-S + GFSA | 12 | 22.0M | 4.57G | 614.54 | 1926.0$s$ |
| | DeiT-S | 24 | 43.3M | 9.09G | 423.68 | 3341.1$s$ |
| | DeiT-S + GFSA | 24 | 43.3M | 9.10G | 314.75 | 3480.3$s$ |
| CaiT | CaiT-S | 24 | 46.9M | 9.34G | 574.66 | 8575.8$s$ |
| | CaiT-S + GFSA | 24 | 47.0M | 9.34G | 406.96 | 8627.5$s$ |
| Swin | Swin-S | 24 | 49.6M | 8.75G | 912.38 | 1897.2$s$ |
| | Swin-S + GFSA | 24 | 49.6M | 8.75G | 714.60 | 1970.0$s$ |

Table 21: Training time (seconds per epoch) on graph classficiation tasks

| Method | ZINC | PCQM4M | PCQM4Mv2 |
|---|---|---|---|
| Graphormer (Ying et al., 2021) | 9$s$ | 740$s$ | 817$s$ |
| Graphormer + GFSA | 9$s$ | 896$s$ | 955$s$ |

Table 22: Training time (seconds per epoch) on LibriSpeech datasets

| Method | LibriSpeech 100h | LibriSpeech 960h |
|---|---|---|
| Transformer | 190.5$s$ | 3049.3$s$ |
| Transformer + GFSA | 191.6$s$ | 3398.4$s$ |
| Branchformer (Peng et al., 2022) | 248.5$s$ | 4990.1$s$ |
| Branchformer + GFSA | 254.3$s$ | 4999.3$s$ |

Table 23: Training time (seconds per epoch) on the code defect prediction task

| Method | Runtime |
|---|---|
| RoBERTa (Liu et al., 2019) | 543.96$s$ |
| RoBERTa + GFSA | 537.79$s$ |
| CodeBERT (Feng et al., 2020) | 555.28$s$ |
| CodeBERT + GFSA | 561.43$s$ |
| PLBART (Ahmad et al., 2021) | 467.80$s$ |
| PLBART + GFSA | 470.19$s$ |
| CodeT5-small (Wang et al., 2021b) | 301.11$s$ |
| CodeT5-small + GFSA | 309.04$s$ |
| CodeT5-base (Wang et al., 2021b) | 362.28$s$ |
| CodeT5-base + GFSA | 373.22$s$ |

# M    INFERENCE TIME

We report the inference time of various methods with GFSA in Tables 24 to 29.

Table 24: Inference time on GLUE tasks. $s$ denotes the abbreviation for second. **Avg** denotes the average training time across all tasks.

| Datasets | #Params | CoLA | SST-2 | MRPC | QQP | STS-B | MNLI-m/mm | QNLI | RTE | **Avg** |
|---|---|---|---|---|---|---|---|---|---|---|
| BERT$_{BASE}$ (Devlin et al., 2019) | 110M | 1.0$s$ | 1.4$s$ | 1.2$s$ | 48.7$s$ | 1.9$s$ | 15.5$s$ | 10.1$s$ | 1.2$s$ | 10.0$s$ |
| BERT$_{BASE}$ + GFSA | 110M | 1.1$s$ | 1.4$s$ | 1.2$s$ | 52.3$s$ | 2.0$s$ | 16.8$s$ | 11.0$s$ | 1.3$s$ | 11.0$s$ |
| ALBERT$_{BASE}$ (Lan et al., 2019) | 11M | 1.1$s$ | 1.6$s$ | 1.4$s$ | 58.4$s$ | 2.2$s$ | 18.4$s$ | 12.1$s$ | 1.3$s$ | 12.0$s$ |
| ALBERT$_{BASE}$ + GFSA | 11M | 1.2$s$ | 1.7$s$ | 1.4$s$ | 62.1$s$ | 2.3$s$ | 19.7$s$ | 13.1$s$ | 1.4$s$ | 13.0$s$ |
| RoBERTa$_{BASE}$ (Liu et al., 2019) | 125M | 1.0$s$ | 1.4$s$ | 1.1$s$ | 47.0$s$ | 1.9$s$ | 15.0$s$ | 9.9$s$ | 1.2$s$ | 10.0$s$ |
| RoBERTa$_{BASE}$ + GFSA | 125M | 1.1$s$ | 1.4$s$ | 1.2$s$ | 50.4$s$ | 2.0$s$ | 16.3$s$ | 10.8$s$ | 1.2$s$ | 11.0$s$ |

Table 25: Inference time on causal language modeling tasks.

| Method | #Params | PTB | WikiText-2 | WikiText-103 | **Avg** |
|---|---|---|---|---|---|
| GPT2 (Radford et al., 2019) | 117M | 3.2$s$ | 7.4$s$ | 7.4$s$ | 6.0$s$ |
| GPT2 + GFSA | 117M | 5.5$s$ | 12.9$s$ | 12.9$s$ | 10.4$s$ |

Table 26: Inference time on ImageNet-1k

| Backbone | Method | #Layers | Inference Time |
|---|---|---|---|
| DeiT | DeiT-S | 12 | 52$s$ |
| | DeiT-S + GFSA | 12 | 53$s$ |
| | DeiT-S | 24 | 68$s$ |
| | DeiT-S + GFSA | 24 | 69$s$ |
| CaiT | CaiT-S | 24 | 105$s$ |
| | CaiT-S + GFSA | 24 | 107$s$ |
| Swin | Swin-S | 24 | 17$s$ |
| | Swin-S + GFSA | 24 | 17$s$ |

Table 27: Inference time on graph classficiation tasks

| Method | ZINC | PCQM4M | PCQM4Mv2 |
|---|---|---|---|
| Graphormer (Ying et al., 2021) | $8s$ | $99s$ | $31s$ |
| Graphormer + GFSA | $8s$ | $117s$ | $29s$ |

Table 28: Inference time on LibriSpeech datasets

| Method | LibriSpeech 100h | LibriSpeech 960h |
|---|---|---|
| Transformer | $328.1s$ | $323.7s$ |
| Transformer + GFSA | $329.5s$ | $343.3s$ |
| Branchformer (Peng et al., 2022) | $299.4s$ | $328.7s$ |
| Branchformer + GFSA | $305.5s$ | $354.1s$ |

Table 29: Inference time on the code defect prediction task

| Method | Inference Time |
|---|---|
| RoBERTa (Liu et al., 2019) | $22.4s$ |
| RoBERTa + GFSA | $23.9s$ |
| CodeBERT (Feng et al., 2020) | $23.8s$ |
| CodeBERT + GFSA | $24.1s$ |
| PLBART (Ahmad et al., 2021) | $37.7s$ |
| PLBART + GFSA | $39.3s$ |
| CodeT5-small (Wang et al., 2021b) | $78.2s$ |
| CodeT5-small + GFSA | $82.5s$ |
| CodeT5-base (Wang et al., 2021b) | $83.2s$ |
| CodeT5-base + GFSA | $88.5s$ |

## N  FREQUENCY ANALYSES IN THE SINGULAR VALUE DOMAIN

Graph signal processing (GSP) (Sandryhaila & Moura, 2013; 2014) can be understood as a generalized concept of DSP — in other words, DSP is a special case of GSP where a *line graph with $n$ nodes* is used and therefore, the graph Fourier transform (GFT) of the line graph is identical to the discrete Fourier transform.

In the definition of GFT, we assume that the graph shift operator (GSO) $S$ is diagonalizable. Considering the eigendecomposition of the GSO $S = V^{\mathsf{T}}\Lambda V$ with eigenvector $V$, we can write the graph filter output as follows:

$$y = \sum_{k=0}^{K} w_k S^k x = \sum_{k=0}^{K} V^{\mathsf{T}} w_k \Lambda^k V x = V^{\mathsf{T}}\Big(\sum_{k=0}^{K} w_k \Lambda^k\Big) V x = V^{\mathsf{T}} g(\Lambda) V x, \qquad (15)$$

where $x \in \mathbb{R}^n$ is a 1-dimensional graph signal, $\Lambda$ is a diagonal matrix with eigenvalues, and $w_k \in [-\infty, \infty]$ is a coefficient.

However, one can use the singular value decomposition, $S = V^{\mathsf{T}}\Lambda U$, when the GSO is not diagonalizable but symmetrically normalized instead of the eigendecomposition (Maskey et al., 2023). Both the singular value decomposition and the eigendecomposition project the original signal onto a set of basis, but they use different basis sets. In the singular value decomposition, we sort the set of basis in ascending order of their eigenvalues, i.e., $\Lambda$, and perform frequency domain-like analyses (Zou et al., 2022; Maskey et al., 2023).

Since the self-attention matrix's row-wise sum is always 1, the following is the case: $\bar{A} = D^{-1}A = \frac{1}{N}A$, where $N$ is the number of tokens. In [1], they define the following symmetrically normalized adjacency matrix (SNA): $D_{in}^{-1/2}AD_{out}^{-1/2}$. Since the degree of every node is $N$ in the self-attention matrix, the following is the case: $D_{in}^{-1/2}AD_{out}^{-1/2} = D^{-1/2}AD^{-1/2} = \frac{1}{\sqrt{N}}A\frac{1}{\sqrt{N}} = \frac{1}{N}A = \bar{A}$. Therefore, the self-attention matrix is a special case of SNAs.

## O  MATRIX POLYNOMIAL VS. GRAPH FOURIER TRANSFORM

There are two paradigms of implementing graph filters: i) matrix polynomial which does not require diagonalizability, ii) graph Fourier transform which uses the eigendecomposition for diagonalizable adjacency matrices or the Jordan decomposition for non-diagonalizable adjacency matrices.

Those two paradigms have their own weaknesses: i) the matrix polynomial approach requires explicit matrix multiplications, and ii) the graph Fourier transform approach requires expansive spectral decompositions. The matrix polynomial is preferred when the number of matrix multiplications is not many. Otherwise, the graph Fourier transform approach may be better since since the matrix multiplication can be simplified after the eigendecomposition or the Jordan decomposition or the Jordan normal form.

Among those two, we use the first matrix polynomial approach with only three non-zero coefficients $\{w_0, w_1, w_K\}$ since it does not require the complicated spectral decomposition. Since we do not rely on any explicit spectral decomposition but on the matrix polynomial, any adjacency matrix can be used.

## P  APPROXIMATION OF $\bar{A}^K$ IN EQUATION 4

The general formulation of first-order taylor approximation at point $a$ is as follows:

$$f(x) \simeq f(a) + f'(a)(x - a). \qquad (16)$$

We approximate $f(K) = \bar{A}^K$ with first-order taylor approximation at point $a = 1$:

$$f(K) = \bar{A}^K \simeq f(1) + f'(1)(K - 1). \qquad (17)$$

Inspired by Brouwer et al. (2019), which approximates the derivative of hidden states of RNNs as the difference between hidden states, we effectively approximate $f'(1)$, the derivative of $\bar{A}^K$

estimated at the position of $K = 1$, with the difference term as $f(2) - f(1) = \bar{A}^2 - \bar{A}^1$. Then the approximated $\bar{A}^K$ becomes as follows:

$$f(K) = \bar{A}^K \simeq f(1) + (\bar{A}^2 - \bar{A})(K - 1) = \bar{A} + (K - 1)(\bar{A}^2 - \bar{A}). \tag{18}$$

This approximation has two advantages: 1) The approximation for $\bar{A}^K$ with $\bar{A}$ and $\bar{A}^2$ provides a simpler computation that can significantly reduce the required computational resources and time (see theoretical analysis in section 3.2). 2) When approximating derivative of $\bar{A}^K$ with difference term for graph filter design, our filter can reduce to GPR-GNN and GREAD, two popular GNN methods preventing the oversmoothing for graph convolutional networks, are special cases of our design (see discussion in section 3.3). In other words, our method is more general than those two GNNs.

