# OpenReview forum: "Graph Convolutions Enrich the Self-Attention in Transformers!"
_ICLR.cc/2024/Conference — Submitted to ICLR 2024_

### Official Review · Reviewer_kKdF · 2023-10-28

**Soundness:** 2 fair
**Presentation:** 3 good
**Contribution:** 2 fair
**Rating:** 5
**Confidence:** 4

**Summary:**

Recent studies unveil that, much like GNNs, Transformers suffer from the over-smoothing issue, where an increase in depth does not consistently lead to enhanced performance. More disconcerting is the finding that a Transformer with a significantly deeper layer does not outperform a shallower one. Stemming from these observations, the authors interpret self-attention as a form of graph convolution. Viewing through the perspective of graph signal processing, they propose Graph Filter-based Self-Attention (GFSA), where the attention matrix is processed through a trinomial-based graph filter. The experimental results affirm that GFSA substantially elevates the performance of Transformers across various domains.

**Strengths:**

1. This paper encompasses extensive experiments covering a variety of domains including language models, vision Transformers, automatic speech recognition, graph Transformers, and code classifications.

**Weaknesses:**

1. The distinction between self-attention and (directed) graph convolution lies in the requirement for the graph shift operator in (directed) graph convolution to be diagonalizable, a constraint absent in the attention matrix in self-attention. In a directed graph, each signal space -- to which each column of input features belongs -- should remain invariant to filtering [1]. In other words, the signal space housing each column of output vectors should be the same as that of the corresponding input vectors. The sole condition fulfilling this scenario is that the graph shift operator has to be diagonalizable. Hence, unless the authors can demonstrate that the attention matrix in each GFSA is diagonalizable, employing graph signal processing as the theoretical underpinning for GFSA is inappropriate.
2. The elucidation provided on how GFSA mitigates the over-smoothing issue lacks adequacy. The authors need to furnish a more comprehensive illustration, accompanied by meticulous formulation derivations and rigorous proofs, to substantiate their claims.
3. The time complexity of GFSA exceeds that of both single-head and multi-head self-attention. Specifically, in scenarios with a large number of tokens, the computation of $\mathbf{A}^{2}$ will demand a substantial amount of computational resources.

[1] A. Sandryhaila and J. M. F. Moura, "Discrete Signal Processing on Graphs," in IEEE Transactions on Signal Processing, vol. 61, no. 7, pp. 1644-1656, April1, 2013, doi: 10.1109/TSP.2013.2238935.

**Questions:**

1. In (directed) graph convolution, the eigenvalues of the graph shift operator are filtered in accordance with the principles of graph signal processing. Assuming the graph filter can be directly applied to the attention matrix, which segment of the attention matrix is subjected to filtering?

---

> ### Author Response · Authors · 2023-11-16
>
> Thanks for your time reading our paper and leaving insightful comments. We uploaded our revised paper, where changes are highlighted in red.
>
> ---
>
> **Q1.Hence, unless the authors can demonstrate that the attention matrix in each GFSA is diagonalizable, employing graph signal processing as the theoretical underpinning for GFSA is inappropriate.**
>
> For our reply to this question, please refer to [Q2.](https://openreview.net/forum?id=poFAoivHQk&noteId=ruiKJRrpJX) of our general response.
>
> ---
>
> **Q2. The elucidation provided on how GFSA mitigates the over-smoothing issue lacks adequacy.**
>
> For our reply to this question, please refer to [Q1.](https://openreview.net/forum?id=poFAoivHQk&noteId=ruiKJRrpJX) and [Q3.](https://openreview.net/forum?id=poFAoivHQk&noteId=ozboZFBU0m) of our general response.
>
> ---
>
> **Q3. The time complexity of GFSA exceeds that of both single-head and multi-head self-attention. Specifically, in scenarios with a large number of tokens, the computation of  will demand a substantial amount of computational resources.**
>
> The approximation for $\bar{A}^K$ with $A$ and $\bar{A}^2$ provides a simpler computation that can significantly reduce the required computational resources and time.  Nevertheless, the computational resources may still exist due to matrix multiplication that calculates $\bar{A}^2$. However, in Tables 18 to 23 in Appendix L, the increases of empirical training time are trivial in most cases.
>
> Also, to handle a large number of tokens, many studies have proposed Sparse Transformers [1,2,3,4] to reduce computational and memory complexities while maintaining performance by treating attention as a sparse graph. For future work, we hope to develop technical approaches to combine our model with sparse attention.
>
>
> > [1] Manzil Zaheer et al. "Big bird: Transformers for longer sequences." NeurIPS, 2020
> >
> > [2] Iz Beltagy, Matthew E. Peters, and Arman Cohan. "Longformer: The long-document transformer." arXiv preprint arXiv:2004.05150, 2020
> >
> > [3]  Nikita Kitaev, Lukasz Kaiser, and Anselm Levskaya. "Reformer: The Efficient Transformer." In International Conference on Learning Representations, 2020.
> >
> > [4] Apoorv Vyas, Angelos Katharopoulos, and François Fleuret. "Fast transformers with clustered attention." NeurIPS, 2020.
>
> ---
>
> **Q4. In (directed) graph convolution, the eigenvalues of the graph shift operator are filtered in accordance with the principles of graph signal processing. Assuming the graph filter can be directly applied to the attention matrix, which segment of the attention matrix is subjected to filtering?**
>
> For our reply to this question, please refer to [Q4.](https://openreview.net/forum?id=poFAoivHQk&noteId=ozboZFBU0m) of our general response.
>
> ---
> Thank you again for taking the time to review our work. If you still have any remaining concerns after our responses, we will do our best to answer them.

---

> ### Comment · Reviewer_kKdF · 2023-11-17
> **Response to Authors (1/4)**
>
> I recommend rejecting this paper. Before outlining my reasons, I would like to clarify certain concepts in Graph Signal Processing, Directed Graph Signal Processing, and Graph Neural Networks. This clarification is intended to aid readers in better comprehending the arguments I present.
>
> 1. graph shift operator: Graph signal processing (GSP) primarily focuses on the analysis of unsigned undirected graphs [1]. A graph shift operator (GSO) [3,4,5], denoted as $\mathbf{S} \in \mathbb{R}^{n \times n}$, is a matrix that defines how a graph signal transitions from one node to its neighboring nodes, based on the graph topology. In GSP, it is customary to employ a (normalized) adjacency matrix or Laplacian matrix as the GSO [3,4,5]. In Graph Neural Networks (GNNs), a normalized self-looped adjacency matrix or Laplacian matrix is typically employed as the GSO, particularly since the introduction of the so-called renormalization trick in GCN [10]. In GSP, a GSO is symmetric, as the adjacency matrix of an unsigned undirected graph is symmetric.
>
> 2. polynomial-based graph filter: In GSP, the concept of a graph filter, denoted as $\mathcal{H}\_{\mathbf{\Theta}}(\mathbf{S}) \in \mathbb{R}^{n \times n}$, is interpreted as a function of the GSO [3,4,5]. It is evident that a GSO represents a special case of a graph filter. The polynomial-based graph filter is defined as $\mathcal{H}\_{\mathbf{\Theta}}(\mathbf{S}) = \sum^{K}\_{k=0}{{\theta}\_{k}\mathbf{S}^{k}}$, where ${\theta}\_{k}$ is the corresponding coefficient.
>
> 3. graph Fourier transform: In essence, the graph Fourier transform (GFT) is a type of the discrete orthogonal transform [3,4,5]. Owing to the symmetry of the GSO, its eigenvector matrix is orthogonal. Consequently, the discrete orthogonal basis of the GFT is derived from the inverse of the eigenvector matrix of the GSO. For a given graph signal $\mathbf{X}$, the GFT is expressed as $\hat{\mathbf{X}} = \mathbf{U}^{-1}\mathbf{X}$, while the inverse graph Fourier transform (IGFT) is represented as $\mathbf{X} = \mathbf{U}\hat{\mathbf{X}}$.
>
> 4. graph convolution: In GNNs, a graph convolution is represented as $\mathrm{GC}(\mathbf{S},\mathbf{X}) = \mathcal{H}\_{\mathbf{\Theta}}(\mathbf{S})\mathbf{X}\mathbf{W} = \sum^{K}\_{k=0}{{\theta}\_{k}\mathbf{S}^{k}}\mathbf{X}\mathbf{W}$. Taking GCN [10] as an instance, in GCN, the GSO is a self-looped normalized adjacency matrix, defined as $\widetilde{\mathcal{A}} = \widetilde{\mathbf{D}}^{-0.5}(\mathbf{A+\mathbf{I}\_{n}})\widetilde{\mathbf{D}}^{-0.5}$, where $\widetilde{\mathbf{D}}\_{u,u} = \sum\_{v}(\mathbf{A}+\mathbf{I}\_{n})\_{u,v}$. The first order of the Chebyshev polynomials of the first kind is adopted, i.e., $\mathrm{GC}(\widetilde{\mathcal{A}},\mathbf{X})\_{\text{GCN}} = \widetilde{\mathcal{A}}\mathbf{X}\mathbf{W}$. In a graph convolution, the graph signal is the linear projection of input feature matrix $\mathbf{X}\mathbf{W}$. Since the adjacency matrix of an unsigned undirected graph is symmetric, the GSO is also symmetric, and the eigenvector matrix of the GSO is orthogonal. Consequently, the graph convolution can be represented as $\mathrm{GC}(\mathbf{S},\mathbf{X}) = \sum^{K}\_{k=0}{{\theta}\_{k}\mathbf{S}^{k}}\mathbf{X}\mathbf{W} = \mathbf{U}\left(\sum\_{k=0}^{K}\theta_{k}\mathbf{\Lambda}^{k}\right)\mathbf{U}^{-1}\mathbf{X}\mathbf{W}$. Here, $\mathbf{U}^{-1}\mathbf{X}\mathbf{W}$ represents the GFT of the graph signal. It is
> clear that the eigenvalues of the GSO are filtered by the graph filter. Furthermore, the graph signal undergoes a GFT, after which it is multiplied by the filter eigenvalue matrix. Following this matrix multiplication, it passes through the IGFT.

---

> ### Comment · Reviewer_kKdF · 2023-11-17
> **Response to Authors (2/4)**
>
> 5. directed graph signal processing: For a directed graph, the adjacency matrix is asymmetric. Consequently, a directed graph signal operator (DGSO) is non-diagonalizable. In directed graph signal processing (DGSP), each signal space -- to which each column of input features belongs -- should remain invariant to filtering [3,4,5,6,7]. In other words, the signal space housing each column of output vectors should be the same as that of the corresponding input vectors. The sole condition fulfilling this scenario is that the DGSO has to be diagonalizable. To meet this condition, there are two common approaches. The first is defining a normal DGSO, such as Chung's digraph Laplacian [12] or magnetic Laplacian [13,14]. The second approach involves constructing a normal DGSO. For this method, one can apply the Jordan composition [3,4,5] or Schur decomposition [9] to the non-normal DGSO to reorganize it into a normal DGSO. Taking the Jordan composition to a non-normal DGSO, denoted as $\mathbf{S}\_{\text{nn}} \in \mathbb{R}^{n \times n}$, as an instance, it can be decomposed into $\mathbf{V}\mathbf{J}\mathbf{V}^{-1}$, where $\mathbf{V}^{-1}$ is the generalized eigenvector matrix of $\mathbf{S}\_{\text{nn}}$, and $\mathbf{J}$ is a Jordan canonical form matrix with eigenvalues on its diagonal. By retaining the diagonal elements in $\mathbf{J}$, the eigenvalue matrix $\mathbf{\Lambda}$ of $\mathbf{S}\_{\text{nn}}$ is obtained. As a result, a normal DGSO can be constructed as $\mathbf{S} = \mathbf{V}\mathbf{\Lambda}\mathbf{V}^{-1}$. Finally, the directed graph Fourier transform (DGFT) can be expressed as $\hat{\mathbf{X}} = \mathbf{V}^{-1}\mathbf{X}$, and the inverse directed graph Fourier transform (IDGFT) can be represented as $\mathbf{X} = \mathbf{V}\hat{\mathbf{X}}$.
>
> 6. self-attention and (directed) graph convolution: In self-attention, the attention matrix $\mathbf{A} = \mathrm{Softmax}(\frac{\mathbf{X}\mathbf{W}\_{\text{Q}}\mathbf{W}\_{\text{K}}^{\mathrm{T}}\mathbf{X}^{\mathrm{T}}}{\sqrt{d}_{k}})$ represents the relationship between the query and the key matrix. When $\mathbf{W}\_{\text{Q}} = \mathbf{W}\_{\text{K}}$, the attention matrix becomes symmetric, indicating that the unidirectional self-attention acts as a graph convolution. As a result, GSP can be used to analyze the unidirectional self-attention. However, when $\mathbf{W}\_{\text{Q}} \neq \mathbf{W}\_{\text{K}}$, the attention matrix may not be diagonalizable. In this case, the self-attention does not act as a directed graph convolution. Consequently, DGSP cannot be directly applied for analysis.

---

> ### Comment · Reviewer_kKdF · 2023-11-17
> **Response to Authors (3/4)**
>
> First, the authors introduce a polynomial-based filter to the attention matrix, grounding its explanation in GSP theory. However, this approach encounters issues when $\mathbf{W}\_{\text{Q}} \neq \mathbf{W}\_{\text{K}}$, as neither GSP nor DGSP are applicable for analyzing this filter. Notably, neither eigenvalues nor singular values of the attention matrix are filtered. This is why I assert that the authors employ GSP to analyze the filter proposed in the paper is inappropriate.
>
> Second, the revised version paper discusses two approaches for implementing graph filters: matrix polynomial and graph Fourier transform. Yet, the authors seem to conflate graph filters with the (directed) graph Fourier transform. If the authors persist in incorporating GSP or DGSP for filter analysis, I strongly recommend exploring options like the unidirectional self-attention, proving the attention matrix is diagonalizable when $\mathbf{W}\_{\text{Q}} \neq \mathbf{W}\_{\text{K}}$, adopting a normal DGSO such as the magnetic Laplacian, reconstruct a diagonalizable DGSO via the Jordan decomposition, or proposing a variant of total variation.
>
> Third, the authors claim that the proposed filter can be reduced to GPR-GNN [11], aiding GFSA in addressing the over-smoothing issue. The complexity of over-smoothing issue is acknowledged. There is substantial theoretical support in GPR-GNN, demonstrating that GPR-GNN can alleviate the over-smoothing issue, even as the order of the polynomial tends towards infinity. Conversely, this paper lacks substantial theory to prove the ability of GFSA to counter over-smoothing. Moreover, the theoretical support in GPR-GNN analyzes filtered eigenvalues in a monomial-based learnable graph filter, a contrast to the eigenvalues or the singular values of the attention matrix is not filtered in the proposed filter. Furthermore, there is scant experimental evidence to support GFSA can alleviate the over-smoothing. A straightforward method to demonstrate counter-over-smoothing capability is by showing that the metric used to judge the model's performance does not decline, which the paper fails to do. For instance, Table 12 shows a significant performance drop in DeiT-T with GFSA from 16-layer to 24-layer on both ImageNet-100 and ImageNet-1k, suggesting an over-smoothing problem that contradicts the authors' claims.
>
> Fourth, the authors overlooks the significant time complexity of GFSA, especially when calculating $\mathbf{A}^{2}$, which has a time complexity of $\mathcal{O}(n^{3})$. This is substantially higher compared to multi-head self-attention's $\mathcal{O}(n^{2})$, with GFSA offering only marginal performance improvement at a substantial computational expense.
>
> In conclusion, considering these four aspects, I advise against accepting this paper.

---

> ### Comment · Reviewer_kKdF · 2023-11-17
> **Response to Authors (4/4)**
>
> # Rerference
> [1] Shuman, D. I., Narang, S. K., Frossard, P., Ortega, A., & Vandergheynst, P. (2013). The Emerging Field of Signal Processing on Graphs: Extending High-Dimensional Data Analysis to Networks and Other Irregular Domains. IEEE Signal Processing Magazine, 30(3), 83–98.
>
> [2] Marques, A. G., Segarra, S., & Mateos, G. (2020). Signal Processing on Directed Graphs: The Role of Edge Directionality When Processing and Learning From Network Data. IEEE Signal Processing Magazine, 37(6), 99–116.
>
> [3] Sandryhaila, A., & Moura, J. M. F. (2013). Discrete signal processing on graphs: Graph fourier transform. 2013 IEEE International Conference on Acoustics, Speech and Signal Processing, 6167–6170.
>
> [4] Sandryhaila, A., & Moura, J. M. F. (2013). Discrete Signal Processing on Graphs. IEEE Transactions on Signal Processing, 61(7), 1644–1656.
>
> [5] Sandryhaila, A., & Moura, J. M. F. (2014). Discrete Signal Processing on Graphs: Frequency Analysis. IEEE Transactions on Signal Processing, 62(12), 3042–3054.
>
> [6] Singh, R., Chakraborty, A., & Manoj, B. S. (2016). Graph Fourier transform based on directed Laplacian. 2016 International Conference on Signal Processing and Communications (SPCOM), 1–5.
>
> [7] Deri, J. A., & Moura, J. M. F. (2017). Spectral Projector-Based Graph Fourier Transforms. IEEE Journal of Selected Topics in Signal Processing, 11(6), 785–795.
>
> [8] Domingos, J., & Moura, J. M. F. (2020). Graph Fourier Transform: A Stable Approximation. IEEE Transactions on Signal Processing, 68, 4422–4437.
>
> [9] Barrufet, J., & Ortega, A. (2021). Orthogonal Transforms for Signals on Directed Graphs. arXiv Preprint arXiv: Arxiv-2110. 08364.
>
> [10] Kipf, T. N., & Welling, M. (2017). Semi-Supervised Classification with Graph Convolutional Networks. International Conference on Learning Representations.
>
> [11] Chien, E., Peng, J., Li, P., & Milenkovic, O. (2021). Adaptive Universal Generalized PageRank Graph Neural Network. International Conference on Learning Representations.
>
> [12] Chung, F. (04 2005). Laplacians and the Cheeger Inequality for Directed Graphs. Annals of Combinatorics, 9(1), 1–19.
>
> [13] Fanuel, M., Alaíz, C. M., & Suykens, J. A. K. (02 2017). Magnetic eigenmaps for community detection in directed networks. Phys. Rev. E, 95, 022302.
>
> [14] Fanuel, M., Alaíz, C. M., Fernández, Á., & Suykens, J. A. K. (2018). Magnetic Eigenmaps for the visualization of directed networks. Applied and Computational Harmonic Analysis, 44(1), 189–199.

---

> > ### Author Response · Authors · 2023-11-17
> > **Responce to Reviewer (2/2)**
> >
> > **Q3. there is scant experimental evidence to support GFSA can alleviate the over-smoothing. Table 12 shows a significant performance drop in DeiT-T with GFSA from 16-layer to 24-layer on both ImageNet-100 and ImageNet-1k, suggesting an over-smoothing problem that contradicts the authors' claims.**
> >
> > Table 12 shows the accuracy at each layer in the best hyperparameter $K$, showing that GFSA boosts the performance of DeiT in all layers. As can be seen in Table 13 which is a sensitivity study on $K$, in ImageNet-100, except when $K$ is 4, performance improves even as the layers become deeper at the same $K$. Also, in ImageNet-1K, you can see that performance does not drop significantly even when layers go deeper from 16 to 24, except when $K$ is 2.
> >
> > Also, the alleviating of oversmoothing is evident in Figure 2 in Introduction and Figures 4 and 5 in Appendix D.
> >
> > 1. As shown in Fig. 2 (a), DeiT has a near-zero magnitude for the high frequencies, which is characteristic of a low-frequency filter and is likely to result in oversmoothing when applied multiple times. On the other hand, DeiT with GFSA has a various frequency information, including high frequency signals, and can alleviate oversmoothing.
> >
> > 2. The higher similarity of feature vector as the layers of the model get deeper is related to the oversmoothing problem. As can be seen in Fig. 2 (b), the cosine similarity increases rapidly as layers go deeper, while DeiT plugged with GFSA does not.
> >
> > 3. If the distribution of singular values of features in the last block is dominated by values close to zero, dimensional collapse occurs. As can be seen in Fig. 2 (c), using GFSA can alleviate this phenomenon.
> >
> >
> > **Q4. Time complexity of $\mathcal{O}(n^3)$ is substantially higher compared to multi-head self-attention's $\mathcal{O}(n^2)$, with GFSA offering only marginal performance improvement at a substantial computational expense.**
> >
> >
> > From the results in Table 3, the performance improvement of our GFSA on ImageNet-1k is not marginal at all. 12-layer DeiT+GFSA outperforms 24-layer DeiT, which can benefit from runtime depending on the number of layers.
> > For example, as shown in Table 20, 24-layer DeiT takes about 50 minutes per epoch, and 12-layer DeiT+GFSA takes about 30 minutes per epoch. However, 12-layer DeiT+GFSA outperforms 24-layer DeiT.
> >
> > Referring to the results of DeiT, the effectiveness of GFSA can significantly reduce the empirical runtime with comparable or improved performance and resolve your concerns about the time complexity of our GFSA.
> >
> > We hope our response addresses your concerns.

---

> ### Author Response · Authors · 2023-11-17
> **Response to Reviewer (1/2)**
>
> **Q1-2. Your concerns on directed graph signal processing**
>
> You insist that graph signal processing is not possible for directed graphs, which is not true. For instance, Fourier transform is a special case of graph Fourier transform where a directed ring graph is used --- see the graph signal exmaple in the second page of the following link.
>
> https://web.media.mit.edu/~xdong/teaching/aims/lab/MT19/AIMS_CDT_SP_MT19_Lab2.pdf
>
> Since you said that the main problem in our paper is that we used directed graphs and graph signal processing cannot be applied in such cases, your review decision in imprecise. I also cite the following article which discusses i) graph signal processing for directed graphs, and ii) graph signal processing for non-diagonalizable cases.
>
> https://www.hajim.rochester.edu/ece/sites/gmateos/pubs/dgft/DGFT_SPMAG.pdf
>
>
> We do not say that we diagonalize the self-attention matrix. This implementaion is not frequently used for deep learning since the diagonalization takes non-trivial computation. Instead, we typically use the matrix polynomial approach to implement graph filters. We summarize our points again as follows.
>
> Our proposed GFSA is designed with matrix polynomials. The main advantage of matrix polynomial filters is that they avoid explicit eigen decomposition while retaining arbitrary filtering capabilities [1]. To reiterate, matrix polynomial-based designs do not require diagonalizable adjacency matrices, whereas explicit graph Fourier transforms do. So our GFSA is not bound by inability to diagonalize self-attention that you are concerned about. Furthermore, as is already well known in the graph community, matrix polynomial filters attract considerable attention in recent years [1-12] and are more widely used than explicit graph Fourier transforms.
>
> > [1] Deyu Bo, Xiao Wang, Yang Liu, Yuan Fang, Yawen Li, and Chuan Shi. "A Survey on Spectral Graph Neural Networks." arXiv preprint arXiv:2302.05631, 2023.
> >
> > [2] Antonio Ortega, Pascal Frossard, Jelena Kovačević, José MF Moura, and Pierre Vandergheynst. "Graph signal processing: Overview, challenges, and applications." Proceedings of the IEEE 106, 2018.
> >
> > [3] Michaël Defferrard, Xavier Bresson, and Pierre Vandergheynst. "Convolutional neural networks on graphs with fast localized spectral filtering." NeurIPS, 2016.
> >
> > [4] Eli Chien, Jianhao Peng, Pan Li, and Olgica Milenkovic. "Adaptive Universal Generalized PageRank Graph Neural Network." ICLR, 2021.
> >
> > [5] Mingguo He, Zhewei Wei, and Ji-Rong Wen. "Convolutional neural networks on graphs with Chebyshev approximation, revisited." NeurIPS, 2022.
> >
> > [6] Mingguo He, Zhewei Wei, and Hongteng Xu. "Bernnet: Learning arbitrary graph spectral filters via Bernstein approximation." NeurIPS, 2021.
> >
> > [7] Xiyuan Wang and Muhan Zhang. "How powerful are spectral graph neural networks." ICML, 2022.
> >
> > [8] Qimai Li, Xiaotong Zhang, Han Liu, Quanyu Dai, and Xiao-Ming Wu. "Dimensionwise separable 2-D graph convolution for unsupervised and semi-supervised learning on graphs." KDD, 2021
> >
> > [9] Felix Wu, Amauri Souza, Tianyi Zhang, Christopher Fifty, Tao Yu, and Kilian Weinberger. "Simplifying graph convolutional networks." ICML, 2019.
> >
> > [10] Bryan Perozzi, Rami Al-Rfou, and Steven Skiena. "Deepwalk: Online learning of social representations." KDD, 2014.
> >
> > [11] Aditya Grover and Jure Leskovec. "Node2Vec: Scalable feature learning for networks." KDD, 2016.
> >
> > [12] Zhiqian Chen, Fanglan Chen, Lei Zhang, Taoran Ji, Kaiqun Fu, Liang Zhao, Feng Chen, Lingfei Wu, Charu Aggarwal, and Chang-Tien Lu. "Bridging the gap between spatial and spectral domains: A unified framework for graph neural networks." ACM Computing Surveys, 2021.

---

> ### Comment · Reviewer_kKdF · 2023-11-17
> **Response to Authors**
>
> I'm very glad to briefly review the 14 references mentioned by the author in the comments. Let us analyze these references one by one and explain the points of confusion and errors made by the authors.
>
> 1. In the document available at https://web.media.mit.edu/~xdong/teaching/aims/lab/MT19/AIMS_CDT_SP_MT19_Lab2.pdf, there is a discussion on Erdős–Rényi random graphs and unnormalized/normalized (combinatorial) graph Laplacians. It is important to note that Erdős–Rényi random graphs are typically directed, not undirected. In this document, an Erdős–Rényi random graph is transformed into an undirected graph to derive an unnormalized combinatorial graph Laplacian. In this context, the undirected graph is represented by the symbol $\mathbf{W}$. Furthermore, the document provides a clear explanation of how the GFT converts a graph signal from the vertex domain to the graph spectral domain. Additionally, it discusses the utilization of a heat kernel filter as a low-pass graph filter, demonstrating the process of filtering the eigenvalues of an unnormalized (combinatorial) graph Laplacian.
>
> 2. The document accessible at https://www.hajim.rochester.edu/ece/sites/gmateos/pubs/dgft/DGFT_SPMAG.pdf explicitly states that in directed graph signal processing, the Graph Signal Operator (GSO) ought to be diagonalizable. Furthermore, it notes that in cases where the GSO is not diagonalizable, employing the Jordan decomposition as an alternative approach to the GSO is a viable option.
>
> 3. In "A Survey on Spectral Graph Neural Networks", it is pointed out that the magnetic Laplacian is adopted in MagNet, a GNN designed for directed graphs. Here the magnetic Laplacian is **diagonalizable**.

---

> > ### Author Response · Authors · 2023-11-17
> > **Let us make our discussion simple.**
> >
> > Let me ask you one question. Is it incorrect to construct a matrix polynomial with a directed adjacency matrix?
> >
> > Please forget about all those spectral decomposition-based methods since we do not follow the direction.

---

> ### Comment · Reviewer_kKdF · 2023-11-17
>
> From the perspective of linear algebra, it is OK to construct a matrix polynomial with a directed adjacency matrix. However, it is not permissible to directly apply GSP and DGSP for the analysis of this filter as theoretical support. **More importantly, if the GSO is not diagonalizable, it is neither directed graph convolution nor graph convolution!**

---

> ### Author Response · Authors · 2023-11-17
> **You knowledge is incorrect.**
>
> For the matrix polynomail approach, the diagonalizability does not need to be assumed. You can find it in many books, papers, and blogs. Please visit the following link as an example. See Section 4 Graph polynomial filter.
>
> https://ietresearch.onlinelibrary.wiley.com/doi/10.1049/iet-spr.2016.0700
>
> There is a paper [1] about this topic. Its link is https://ieeexplore.ieee.org/document/8309036.
>
> [1] A. Sakiyama, T. Namiki and Y. Tanaka, "Design of polynomial approximated filters for signals on directed graphs," 2017 IEEE Global Conference on Signal and Information Processing (GlobalSIP), Montreal, QC, Canada, 2017, pp. 633-637, doi: 10.1109/GlobalSIP.2017.8309036.

---

> ### Comment · Reviewer_kKdF · 2023-11-17
>
> OK. Let us dicuss this document which is available at https://ietresearch.onlinelibrary.wiley.com/doi/10.1049/iet-spr.2016.0700 . The document states "In general, $\mathbf{A}^{\mathrm{T}}$ is not exactly diagonalisable and an approximation is used $\mathbf{A}^{\mathrm{T}} \simeq \mathrm{h}(\mathbf{\mathbf{A}}) = \hat{\mathbf{V}}\\hat{\mathbf{D}}\hat{\mathbf{V}}^{-1}$". Here, $\mathrm{h}(\mathbf{\mathbf{A}})$ is **diagonalizable**. In this
> sentence, the authors use a diagonalizable operator to approximate a non-diagonalizable operator. Moreover, the authors use $(\mathbf{I}\_{n} - \mathbf{A})^{\mathrm{T}}(\mathbf{I}\_{n} - \mathbf{A})$ to create a symmetric operator $\mathbf{B}$ to the graph filter.
>
> As for anothor paper which is available at https://ieeexplore.ieee.org/document/8309036. The paper states "We will only consider a
> connected, finite, directed graph with no multiple and non-negative edges, and we also assume that $\mathcal{A}$ is **diagonalizable**."
>
> **As you can see, both two documents support my arguments: The GSO should be diagonalizable. If the GSO is not diagonalizable, you need to find an approach to rebuild a diagonalizable GSO!**
>
> As you mentioned that "For the matrix polynomail approach, the diagonalizability does not need to be assumed. You can find it in many books, papers, and blogs.", you can provide me any books, papers, and blogs that support your arguments. I am open to discussing whether the GSO should be diagonalizable or not.

---

> ### Author Response · Authors · 2023-11-17
> **Response by Author**
>
> Please find the following sentence in https://arxiv.org/pdf/2211.08854.pdf.
>
> In the definition of GFT, we are assuming the GSOS is diagonalizable.While definitions of GFT for nondiagonalizable GSOs exist [22], [54], [55], we hold to the diagonalizability assumption for a consistent and simple exposition.
>
> It says that GFT exists for nondiagonalizable GSOs, but only for ease of discussion, they assume diagonalizable GSOs.
>
> The theory exists for nondiagonalizable GSOs!
>
> In addition, there is no problem in using directed adjacency matrix.

---

> ### Comment · Reviewer_kKdF · 2023-11-17
>
> In "Graph Filters for Signal Processing and Machine Learning on Graphs", it states that "In the definition of GFT, we are assuming the GSO $\mathbf{S}$ is diagonalizable. While definitions of GFT for nondiagonalizable GSOs exist [22], [54], [55], we hold to the diagonalizability assumption for a consistent and simple exposition.", where [22] is A. Sandryhaila and J. M. F. Moura, "Discrete Signal Processing on Graphs: Frequency Analysis," in IEEE Transactions on Signal Processing, vol. 62, no. 12, pp. 3042-3054, June15, 2014, doi: 10.1109/TSP.2014.2321121., [54] is S. Sardellitti, S. Barbarossa and P. D. Lorenzo, "On the Graph Fourier Transform for Directed Graphs," in IEEE Journal of Selected Topics in Signal Processing, vol. 11, no. 6, pp. 796-811, Sept. 2017, doi: 10.1109/JSTSP.2017.2726979., and [55] is R. Shafipour, A. Khodabakhsh, G. Mateos and E. Nikolova, "A digraph fourier transform with spread frequency components," 2017 IEEE Global Conference on Signal and Information Processing (GlobalSIP), Montreal, QC, Canada, 2017, pp. 583-587, doi: 10.1109/GlobalSIP.2017.8309026.
>
> Each paper in [22], [54], [55] proposes a variant of total variation, eliminating the need to transform the GSO into a diagonalizable form. In other words, this represents an alternative approach to handling non-diagonalizable GSOs. **Do the authors propose a variant of total variation in the paper?**

---

> > ### Author Response · Authors · 2023-11-17
> > **Response to Reviewer**
> >
> > We use a matrix polynomial approach. This approach is simple yet effective. See the following one of the most cited papers about graph signal processing: https://users.ece.cmu.edu/~asandryh/papers/icassp13.pdf. Theorem 2 justifes the matrix polynomial approach. Many GNNs use the matrix polynomial approach.
> >
> > BTW, why do you change your statement? You said diagonalizable GSOs are mandatory for GFT.

---

> ### Comment · Reviewer_kKdF · 2023-11-17
>
> First, the paper available in https://users.ece.cmu.edu/~asandryh/papers/icassp13.pdf utilizes a **diagonalizable** GSO, which is not the case in your paper. Second, in this review, I have never used the word 'mandatory' due to my limited English proficiency.

---

> ### Comment · Reviewer_kKdF · 2023-11-17
> **Theoretical Concern**
>
> My core concern is that if we insert a polynomial into GSO without considering whether it needs to be diagonalizable, it could overturn the current theoretical basis of DGSP. This means that for directed graphs or signed directed graphs, we do not need to do any special preprocessing and can directly insert any polynomial for graph representation learning. From this perspective, I must evaluate the paper with great caution.

---

> ### Author Response · Authors · 2023-11-22
> **Please check our last messages.**
>
> Dear Reviewer kKdF,
>
> Thanks for your thorough review to our paper. We uploaded two global messages whose titles are as follows:
>
> 1) We left one global message with a title of "linear, shift-invariant vs. linear, shift-invariant, diagonalizable."
> 2) We added one additional message to the above global message, whose title is "Additional justification."
>
> We believe that our messages will address your concerns.
>
> Thanks,
> Authors

---

> ### Comment · Reviewer_kKdF · 2023-11-22
>
> Dear authors,
>
> Thank you for your responses. I believe that time will be the best judge of this paper's merits. Ultimately, I decided to increase the score.
>
> Best wishes,
>
> Reviewer kKdF

---

### Official Review · Reviewer_sAYQ · 2023-10-29

**Soundness:** 3 good
**Presentation:** 3 good
**Contribution:** 3 good
**Rating:** 6
**Confidence:** 5

**Summary:**

Transformer can be viewed as an attention-based graph neural networks over fully connected graph and thus suffer from over-smoothing problem, that the representations induced by deep layers converge to distinguishable values and further results in performance degradation. Inspired by graph representation learning, the paper introduces high-order neighbor information aggregation in Transformer’s attention layer to relieve over-smoothing. Experimental study demonstrated that the proposed method could improve the performance of Transformer over a variety tasks. The paper is well-written and easy to follow.

**Strengths:**

1.Transformer can be viewed a special case of attention based GNNs. Therefore, introducing graph-filter to enhance Transformer’s attention is promising and effective.
2.Experiments over variant tasks (CV, NLP and Graph, etc.) demonstrated the proposed method can significantly improve the performance of Transformers.

**Weaknesses:**

1.The bottleneck of standard Transformer in practice is its high time/space complexity O(N^2), the proposed method introduce high-order neighbor information aggregation introduces more computation/memory overheads, even with approximate computation.

**Questions:**

1.How to get the pre-trained large language models integrated with GFSA? Training from scratch or initializing with pre-trained language models and fine-tuning with added GFSA?

2.In section 4.6, the paper talked about the overhead of training time. It is better to add the comparison of inference time.

3.In Eq. (6), do we have any constraints over the coefficients w0, w1, wk? For example w0, w1, wk > 0? In standard Transformer, there is a residual connection in self-attention layer. What is the difference or connection between w0 and this residual connection?

Are these learnable coefficients added to all layers? If yes, are they shared across different layers? Similar questions for multi-head attention, are they shared across different heads?

Missed references:
1) Graph Attention Networks, ICLR, 2017
2) Diffusion Improves Graph Learning, NIPS, 2019
3) Multi-hop Attention Graph Neural Networks, IJCAI 2021

---

> ### Author Response · Authors · 2023-11-16
>
> Thanks for your time reading our paper and leaving insightful comments. We uploaded our revised paper, where changes are highlighted in red.
>
> ---
>
> **Q1. The proposed method introduce high-order neighbor information aggregation introduces more computation/memory overheads, even with approximate computation.**
>
> In Tables 18 to 23 in Appendix L, we report the training time of various methods with GFSA. In general, the training time of methods with GFSA is slightly longer than that of existing methods. For example, the Transformer for the automatic speech recognition task increases from 190.5 seconds to 191.6 seconds on Librispeech 100h dataset, as increases of only 1 second. Instead of computing higher-order polynomial terms, our GFSA approximates them, with only a small increase in runtime, which is not very significant.
>
> ---
>
> **Q2. How to get the pre-trained large language models integrated with GFSA?**
>
> Thank you for your question. In our proposed method, we can integrate GFSA with a pre-trained large language model (e.g., GPT2) by initializing with the pre-trained models and fine-tuning them with replaced GFSA. We use a weight of pre-trained models provided by HuggingFace (https://huggingface.co/models).
>
> ---
>
> **Q3. It is better to add the comparison of inference time?**
>
> We appreciate the suggestion. In the tables below, we report a comparison of inference time between Transformers with and without GFSA to provide a more comprehensive analysis of the computational implications of GFSA. We included those results in our revised version.
>
> Our inference time remains the same or has a minor increase as seen in text classification. For example, in the CoLA dataset, BERT$_{\text{BASE}}$ +GFSA only increases by 1 second, or in the MNLI dataset, it increases by 1.3 seconds. However, if the number of tokens exceeds 1000, such as GPT2, the time increases by about 5 seconds, as in WikiText-2. However, considering the performance increase, we believe this increase in inference time is trivial.
>
>
>
> * Inference time on the text classification task
>
>
> | Method | CoLA | SST-2 | MRPC | QQP | STS-B | MNLI-m/mm | QNLI | RTE |
> | -------- | -------- | -------- | -------- | -------- | -------- | -------- | -------- | -------- |
> | BERT$_{\text{BASE}}$ | 1.0s | 1.4s | 1.2s | 48.7s | 1.9s | 15.5s | 10.1s | 1.2s | 10.0s |
> | BERT$_{\text{BASE}}$ + GFSA | 1.1s | 1.4s | 1.2s | 52.3s | 2.0s | 16.8s | 11.0s | 1.3s | 11.0s |
> | ALBERT$_{\text{BASE}}$  | 1.1s | 1.6s | 1.4s | 58.4s | 2.2s | 18.4s | 12.1s | 1.3s | 12.0s |
> | ALBERT$_{\text{BASE}}$  + GFSA | 1.2s | 1.7s | 1.4s | 62.1s | 2.3s | 19.7s | 13.1s | 1.4s | 13.0s |
> | RoBERTa$_{\text{BASE}}$  | 1.0s | 1.4s | 1.1s | 47.0s | 1.9s | 15.0s | 9.9s | 1.2s | 10.0s |
> | RoBERTa$_{\text{BASE}}$  + GFSA | 1.1s | 1.4s | 1.2s | 50.4s | 2.0s | 16.3s | 10.8s | 1.2s | 11.0s |
>
> * Inference time on the language modeling task
>
> | Method | PTB | WikiText-2  | WikiText-103 |
> | -------- | -------- | -------- | -------- |
> | GPT2        | 3.2s     | 7.4s     | 7.4s     |
> | GPT2 + GFSA | 5.5s     | 12.9s     | 12.9s     |
>
>
> * Inference time on the image classification task
>
>
> | Method | #Layer | Inference Time|
> | -------- | -------- | -------- |
> | DeiT-S        | 12 | 52s |
> | DeiT-S + GFSA | 12 | 53s |
> | DeiT-S        | 24 | 68s |
> | DeiT-S + GFSA | 24 | 69s |
> | CaiT-S        | 24 | 105s|
> | CaiT-S + GFSA | 24 | 107s|
> | Swin-S        | 24 | 17s |
> | Swin-S + GFSA | 24 | 17s |
>
> * Inference time on the graph classification task
>
> | Method | ZINC | PCQM4M  | PCQM4Mv2 |
> | -------- | -------- | -------- | -------- |
> | Graphormer        | 8s     | 99s     | 31s     |
> | Graphormer + GFSA | 8s     | 117s     | 39s     |
>
> * Inference time on the speech recognition task
>
> | Method | LibriSpeech 100h | LibriSpeech 960h|
> | -------- | -------- | -------- |
> | Transformer         | 328.1s | 323.7s |
> | Transformer + GFSA  | 329.5s | 343.3s |
> | Branchformer        | 299.4s | 328.7s |
> | Branchformer + GFSA | 305.5s | 354.1s |
>
> * Inference time on the code defect prediction task
>
>
> | Method   | Inference Time |
> | -------- | -------- |
> |RoBERTa             | 22.4s |
> |RoBERTa + GFSA      | 23.9s |
> |CodeBERT            | 23.8s |
> |CodeBERT + GFSA     | 24.1s |
> |PLBART      | 37.7s |
> |PLBART + GFSA       | 39.3s |
> |CodeT5-small        | 78.2s |
> |CodeT5-small + GFSA | 82.5s |
> |CodeT5-base         | 83.2s |
> |CodeT5-base + GFSA  | 88.5s |

---

> ### Author Response · Authors · 2023-11-16
>
> **Q4. Are there any constraints on the coefficients $w_0$, $w_1$, $w_K$? For example $w_0, w_1, w_K > 0$? In a standard Transformer, there is residual connectivity in the self-attention layer. What is the difference or connection between $w_0$ and this residual connection?**
>
> The coefficients $w_0$, $w_1$, and $w_K$ in Eq.(5) are all real numbers without constraints on their values.
> Regarding the connection with the residual connection in the standard Transformer, the role of w0 in our proposed method is similar to the residual connection. The original residual connection adds the input vector $\mathbf{X}$ to the result of multi-head attention. However, $w_0$ in GFSA adjusts the weight of the value vector $\mathbf{X}\mathbf{W}_{val}$ in each head. Therefore, our GFSA has the advantage of being applicable without changing the internal structure of any Transformers except for the self-attention matrix.
>
> ---
>
> **Q5. Are these learnable coefficients added to all layers? If yes, are they shared across different layers? Similar questions for multi-head attention, are they shared across different heads?**
>
> Thank you for raising these questions. The learnable coefficients are applied to all layers in our proposed method. However, they are not shared across different layers. Each layer has its own set of coefficients. Similarly, for multi-head attention, the learnable coefficients are not shared across different heads. Each head has its own set of coefficients.
>
> ---
>
> **Q6. Missed references**
>
> Thanks for your suggestion. We have included the missed references you mentioned in our revised paper.
>
> ---
>
> We sincerely hope our response will lead to a better understanding of our proposed method and could lead to a fair and positive review.

---

> > ### Comment · Reviewer_sAYQ · 2023-11-22
> >
> > Thanks for the authors' response and answer most of the raised questions. I would like to keep my score.

---

> > > ### Author Response · Authors · 2023-11-23
> > >
> > > Dear, Reviewer sAYQ
> > >
> > > Thank you for reading our response.
> > >
> > > We would like to highlight an example where our method can bring the benefit of improving performance and reducing runtime at the same time, with respect to the weakness pointed out by the reviewer.
> > >
> > > From the results in Table 3, 12-layer DeiT+GFSA outperforms 24-layer DeiT, which can benefit from runtime depending on the number of layers. For example, as shown in Table 20, 24-layer DeiT takes about 50 minutes per epoch, and 12-layer DeiT+GFSA takes about 30 minutes per epoch. However, 12-layer DeiT+GFSA outperforms 24-layer DeiT.
> > >
> > > Referring to the results of DeiT, the effectiveness of GFSA can significantly reduce the empirical runtime with comparable or improved performance and resolve your concerns about the computation overhead of our GFSA.
> > >
> > > We would appreciate it if you would consider the above strength for your final decision.
> > >
> > > Sincerely,
> > >
> > > Authors

---

### Official Review · Reviewer_huFG · 2023-11-01

**Soundness:** 3 good
**Presentation:** 3 good
**Contribution:** 2 fair
**Rating:** 3
**Confidence:** 4

**Summary:**

The paper addresses the over-smoothing problem in deep Transformer models, where representations in different layers become too similar and lead to reduced performance. To tackle this, the authors propose graph-filter-based self-attention (GFSA) to preserve diverse features. GFSA is demonstrated to enhance Transformer model performance in various domains, including computer vision, natural language processing, graph pattern classification, speech recognition, and code classification.

**Strengths:**

1. The paper is easy to read.

2. The paper is well-motivated

**Weaknesses:**

1. In the paper, there is no theory that explains why transformers suffer from overs-moothing as well as how the proposed method mitigates the issue.

2. The derivation in the paper lacks cohesion, as seen in equation 5, where the authors do not provide an explanation for the first-order approximation of A-bar^K. I am doubtful about the correctness of this equation.

3. Theorem 3.1 establishes an error bound for the Taylor approximation in Eqn 5. However, the error bound presented in Equation 9 relies on ||A-bar^2 - A-bar||, making this error bound meaningless since it lacks any bound for ||A-bar^2 - A-bar||.

**Questions:**

1. The authors’ argument on adding the term (A-bar^2 - A-bar) (see Eqn. 6) in Section 3.3 helps alleviate the over-smoothing problem is not sound. The claim that “(A-bar^2 − A-bar) captures the difference between 2-hop and 1-hop neighborhood information” and “acts as a filter that emphasizes changes or variations in a local structure” is evidenceless. Graph neural networks (GNNs) typically require multiple hops to aggregate non-local information from distant neighbors, whereas self-attention allows one to attend all tokens simultaneously. This raises the question of why self-attention would need (A-bar^2 - A-bar).

2. When explaining over-smoothing problem in transformers, the authors claim: “This problem is obvious to understand since Transformers’ aggregation methods for value vectors are simply weighted averages”. This argument is weak and vague.

3. The experimental results display inconsistencies. It is challenging to find the 3.3% improvement mentioned in the introduction for the image classification task, as it is not evident in Figure 1 and Table 3. Likewise, the claim of a 1.63% improvement for DeiT-S + GFSA over DeiT-S does not correspond with the results presented in Table 3. Furthermore, the alleged 6.23% improvement in the natural language understanding task cannot be observed when comparing Figure 1 and Table 5.

In summary, I recommend rejecting the paper for two primary reasons. First, the paper's novelty and contribution are lacking. The concept of self-attention as a weighted graph has already been discussed in prior works [1, 2, 3, 4]. Although the authors introduce the idea of viewing self-attention as a graph filter, they fail to provide a solid explanation for how this perspective addresses the issue of over-smoothing in Transformers. Second, the paper's theoretical foundation is weak, and it lacks evidence to explain why Transformers experience over-smoothing within their graph-based framework and how their proposed models effectively resolve this problem. Lastly, I find that the main arguments presented in the paper are not convincing.

References:

[1] Han Shi, Jiahui Gao, Hang Xu, Xiaodan Liang, Zhenguo Li, Lingpeng Kong, Stephen Lee, and James T Kwok. Revisiting over-smoothing in BERT from the perspective of graph. ICLR, 2022.
[2] Yun, Seongjun, Minbyul Jeong, Raehyun Kim, Jaewoo Kang, and Hyunwoo J. Kim. "Graph transformer networks." NeurIPs, 2019.
[3] Wang, Yuxin, Chu-Tak Lee, Qipeng Guo, Zhangyue Yin, Yunhua Zhou, Xuanjing Huang, and Xipeng Qiu. "What dense graph do you need for self-attention?." ICML, 2022.
[4] Velickovic, Petar, Guillem Cucurull, Arantxa Casanova, Adriana Romero, Pietro Lio, and Yoshua Bengio. "Graph attention networks.". ICLR, 2018.

---

> ### Author Response · Authors · 2023-11-16
>
> Thanks for your time reading our paper and leaving insightful comments. We uploaded our revised paper, where changes are highlighted in red.
>
> ---
>
> **Q1. Why transformers suffer from over-smoothing as well as how the proposed method mitigates the issue**
>
> For our reply to this question, please refer to [Q1.](https://openreview.net/forum?id=poFAoivHQk&noteId=ruiKJRrpJX) and [Q3.](https://openreview.net/forum?id=poFAoivHQk&noteId=ozboZFBU0m) of our general response.
>
> ---
>
> **Q2. The derivation in the paper lacks cohesion, as seen in eq. (4), where the authors do not provide an explanation for the first-order approximation of $\bar{A}^K$. I am doubtful about the correctness of this equation.**
>
> The general formulation of first-order Taylor approximation at point $a$ is as follows:
>
> $f(x) \simeq f(a)+f'(a)(x-a)$
>
> We approximate $f(K)=\bar{A}^K$ with first-order taylor approximation at point $a=1$:
>
> $f(K)=\bar{A}^K\simeq f(1)+f'(1)(K-1).$
>
> Inspired by [1], which approximates the derivative of hidden states of RNNs as the difference between hidden states, we effectively approximate $f'(1)$, the derivative of $\bar{A}^K$ estimated at the position of $K=1$, with the difference term, $f(2)-f(1)=\bar{A}^2-\bar{A}^1$. Then the approximated $\bar{A}^K$ becomes as follows:
>
> $f(K) =\bar{A}^K\simeq f(1)+(\bar{A}^2-\bar{A})(K-1) = \bar{A}+(K-1)(\bar{A}^2-\bar{A})$
>
> This approximation has two advantages:
>
> 1) The approximation for $\bar{A}^K$ with $\bar{A}$ and $\bar{A}^2$ provides a simpler computation that can significantly reduce the required computational resources and time (see theoretical analysis in section 3.2).
>
> 2) When approximating derivative of $\bar{A}^K$ with the difference term, GPR-GNN and GREAD, two popular GNN methods preventing the oversmoothing for graph convolutional networks, are special cases of our design (see discussion in Section 3.3). In other words, our method is more general than those two GNNs.
>
> In particular, the second advantage gave us confidence to some degree even before experimenting with our method. We have included the detailed derivation process of the equation in our revised version.
>
> > [1] Edward De Brouwer, et al. "GRU-ODE-Bayes: Continuous modeling of sporadically-observed time series." NeurIPS, 2019.
>
> ---
>
> **Q3. the error bound presented in Eq. (7) relies on $||\bar{A}^2 - \bar{A}||$, making this error bound meaningless since it lacks any bound for $||\bar{A}^2 - \bar{A}||$.**
>
> Thank you for pointing it out, we further develop the error bound in Eq. (8) as follows:
>
> $E_K \leq 2K,$
>
> where $K$ is the order of polynomial.
>
> Since $||\bar{A}||\leq 1$ and $||\bar{A}^2||\leq ||\bar{A}||^2 \leq 1^2=1$, the upper bound of $E_k$ is becomes as follows:
>
>
> $E_k \leq 2+(K-1)||\bar{A}^2-\bar{A}||\leq 2+(K-1)(||\bar{A}^2||+||\bar{A}||) \leq 2+(K-1)(1+1) = 2K.$
>
> We have included this response in our revised version.
>
> ---
>
> **Q4. “Graph neural networks (GNNs) typically require multiple hops to aggregate non-local information from distant neighbors, whereas self-attention allows one to attend all tokens simultaneously. This raises the question of why self-attention would need $\bar{A}^2 - \bar{A}$."**
>
> For our reply to this question, please refer to [Q3.](https://openreview.net/forum?id=poFAoivHQk&noteId=ozboZFBU0m) of our general response.

---

> ### Author Response · Authors · 2023-11-16
>
> **Q5. When explaining over-smoothing problem in Transformers, the authors claim: “This problem is obvious to understand since Transformers’ aggregation methods for value vectors are simply weighted averages”. This argument is weak and vague.**
>
> We apologize for the vagueness of the argument. We corrected the ambiguity in the sentence raised by the reviewer in the revised paper.
>
> ---
>
> **Q6. The experimental results display inconsistencies**
>
> Thanks for pointing it out. We apologize for one error in the performance improvement percentages we wrote in the Introduction section. We also found that the notation of performance improvements in Table 3 could be confusing, so we modified the name of the rightmost column. We have uploaded the corrected version.
>
> ---
>
> **Q7. the concept of self-attention as a weighted graph has already been discussed in prior works [1, 2, 3, 4]**
>
> We appreciate your evaluation and the concerns raised. In addition to the papers mentioned by the reviewer, research using GAT or self-attention (scaled-dot product attention) as an adjacency matrix for graphs is popular and active.
>
> However, our study is different from the research line that experiments on graph tasks by learning the edge weights of the graph from attention. While the concept of self-attention as a weighted graph has been discussed in prior works, our paper extends this idea by introducing the perspective of viewing self-attention as a graph filter. From this perspective, we describe the similarities and differences in the papers mentioned by the reviewer:
>
> - The paper [1], which uses a hierarchical fusion strategy, was inspired by JKNet [5] and used it as is for self-attention of Transformers. On the other hand, we design self-attention from the perspective of a graph filter in graph signal processing.
> - GTN [2] focuses on creating a meta-path adjacency matrix to utilize the graph type of heterophilic nodes. The attention score used by GTN is the attention score for the candidate adjacency matrix when generating the meta-path adjacency matrix, so it is different from the Transformer's self-attention covered in our paper.
> - Hypercube Transformer [3] belongs to the line of research on sparse attention. This model focuses on how dense self-attention should be to reduce complexity and at the same time maintain performance, and does not interpret self-attention from the perspective of a graph filter.
> - GAT [4], the most popular graph neural network, also uses attention scores to consider it as a weighted graph. However, GAT's attention is Bahdanau et al. attention [6] and Vaswani et al. self-attention [7] is different. While GAT is a model for solving the problems of existing spectral and spatial GCNs, we solve the problem of Transformer's self-attention from the perspective of graph signal processing.
>
> > [1] Han Shi, Jiahui Gao, Hang Xu, Xiaodan Liang, Zhenguo Li, Lingpeng Kong, Stephen Lee, and James T Kwok. Revisiting over-smoothing in BERT from the perspective of graph. ICLR, 2022.
> >
> > [2] Seongjun Yun, Minbyul Jeong, Raehyun Kim, Jaewoo Kang, and Hyunwoo J. Kim. "Graph transformer networks." NeurIPS, 2019.
> >
> > [3] Yuxin Wang, Chu-Tak Lee, Qipeng Guo, Zhangyue Yin, Yunhua Zhou, Xuanjing Huang, and Xipeng Qiu. "What dense graph do you need for self-attention?." ICML, 2022.
> >
> > [4] Petar Velickovic, Guillem Cucurull, Arantxa Casanova, Adriana Romero, Pietro Lio, and Yoshua Bengio. "Graph attention networks.". ICLR, 2018.
> >
> > [5] Keyulu Xu, Chengtao Li, Yonglong Tian, Tomohiro Sonobe, Ken-ichi Kawarabayashi, and Stefanie Jegelka. "Representation learning on graphs with jumping knowledge networks." ICML, 2018.
> >
> > [6] Dzmitry Bahdanau, Kyunghyun Cho, and Yoshua Bengio. "Neural machine translation by jointly learning to align and translate." ICLR, 2015.
> >
> > [7] Ashish Vaswani, Noam Shazeer, Niki Parmar, Jakob Uszkoreit, Llion Jones, Aidan N. Gomez, Łukasz Kaiser, and Illia Polosukhin. "Attention is all you need." NeurIPS, 2017.
>
> ---
>
> We sincerely hope our response will lead to a better understanding of our proposed method. Please do let us know if there are any remaining concerns that have not been fully addressed in our responses.

---

> ### Author Response · Authors · 2023-11-22
> **Gentle Reminder**
>
> Dear Reviewer huFG,
>
> Please check our above messages prepared solely for you.
>
> In addition, please check our two global messages in the top of this page. We uploaded two global messages whose titles are as follows:
>
> 1) We left one global message with a title of "linear, shift-invariant vs. linear, shift-invariant, diagonalizable."
> 2) We added one additional message to the above global message, whose title is "Additional justification."
>
> We believe that our messages will address your concerns.
>
> Thanks,
> Authors

---

### Official Review · Reviewer_fbar · 2023-11-01

**Soundness:** 4 excellent
**Presentation:** 4 excellent
**Contribution:** 4 excellent
**Rating:** 10
**Confidence:** 4

**Summary:**

This paper proposes a novel graph filtering approach to enrich the self-attention in Transformer models. The authors show that this improves the performance on 6 different tasks with different types of data. The issue with standard self-attention is that it may oversmooth across different layers (tokens become too similar across layers). In the proposed formulation, the signal for each token is more rich and endowed with both lower and higher frequency information, which mitigates the risk of oversmoothing. The paper is very ambitious in terms of experiments, systematic in its scientific approach, and well-written.

**Strengths:**

* The problem the article addresses is clearly stated (the over-smoothing problem in Transformers.)
* Clear performance improvement is presented on multiple tasks.
* The idea is conceptually very appealing.
* Experiments to ensure that the frequency response is indeed inriched, and that the cosine similarity across layers can be mitigated with GFSA are included in the paper (e.g., Fig 2a-b)
* Runtime and FLOPS is measured and is thoroughly presented.
* The approach is well presented and easy-to-grasp, especially with the generous supplementary.

**Weaknesses:**

* Table 5 is the only table to provide uncertainty on the estimates. If this is for computational reasons, it would be good to comment on, either in the main text or in the appendix. Generally, it's good to comment on how secure your estimates are. (I see in the appendix that you specify the fixed seed you run on for the other experiments -- great, but I still maintain my point.)

**Questions:**

* What do you mean here? "$w_K$ is learned to negative value" (page 5, comparison to GCNs)
* It seems like you are citing Schwartz et al. here, but to me it seems a little strong to claim that "The computations required for machine learning research are doubling every few months". If this were true we would very quickly be in a very dangerous situation (if it's true doubling). If not, is it possible to tone down the statement to 'rapidly growing' or similar -- or adding 'currently doubling'.

### Detailed minor comments:

* Did you mean V instead of U after Eq. 3 (page 3)?
* Page 3: missing capitalization of Equation in multiple places
* "latent representations tend to become similar to each other, leading to a loss of distinctiveness" (I propose to rephrase this sentence, perhaps deleting the second clause as it does not add any new information -- or add 'loss of distinctiveness in the representations'.)
* English: "due to the issues" should be "due to issues"
* "sometimes uniformity among patches or tokens" >> would be nice to be a bit more precise here. English-wise, maybe you could say 'sporadic' instead of sometimes if you want an adjective, or delete 'sometimes' entirely.
* Section 4.4, would it look nicer to write "graph transformers" in plural instead of 'graph transformer'? Both in section title and in text
* I would increase the column width a little bit in Table 3, and put #Layers in plural to be consistent across layers and params.

---

> ### Author Response · Authors · 2023-11-16
>
> We appreciate your recognition of the contribution and value of our work. We are excited and confident about our ideas and verified performance in the experiment section. This is because the frequency response in Figure 2 actually becomes richer and mitigates the cosine similarity.
>
> We uploaded our revised paper, where changes are highlighted in red.
>
> ---
>
> **Q1. Uncertainty on the estimates in Table 5**
>
> We apologize for confusion. We conducted experiments following the experimental environments of Graphormer using 4 different seeds and reported mean values of results. However, as you pointed out, there is an inconsistency in that the uncertainty on the estimations is reported only in Table 5. To address this, we have removed the uncertainty from Table 5. Due to space constraints, we include the results with uncertainty in Tables 14 and 15 in Appendix of the revised version.
>
> ---
>
> **Q2. The meaning of “is learned to negative value” in page 5.**
>
> In GFSA, the coefficient $w_K$ can be learned to have both positive and negative values. The negative value of $w_K$ allows GFSA to extract high-frequency information from the value vectors, which is different from self-attention which can extract only low-frequency information.
>
> ---
>
> **Q3. In the ethical statements section, is it possible to tone down the statement to ‘rapidly growing’ or similar – or add ‘currently doubling’?**
>
> Thank you for the suggestion. We agree that the statement can be toned down to avoid potential misinterpretation. We will revise the expression to convey that the computations required for machine learning research are rapidly growing.
>
> ---
>
> **Minor comments**
>
> All minor comments you raised have been reflected in the revised paper.

---

### Public Comment · ~Xinyi_Wu3 · 2023-11-15
**Oversmoothing is not a trivial problem in Transformers precisely because of attention**

There is one thing that I think the authors need to be more careful with as I was reading the paper. In the introduction, the paper makes the following claim:

> This problem (oversmoothing) is obvious to understand since Transformers’ aggregation methods for
value vectors are simply weighted averages.

This claim is unfortunately not true. If the weights used to calculate the weighted average stayed the same across different layers, then the claim would make sense. However, for attention-based architectures, the aggregation weights change across different layers because they are calculated by attention. Thus, why oversmoothing would occur in attention-based architectures such as Transformers is a nontrivial problem (and has been controversial in the literature).

Please refer to our NeurIPS 2023 paper (https://openreview.net/forum?id=Kg65qieiuB) for more details about why this is a nontrivial phenomenon and a difficult problem to study, and discussion on common misconception about the problem in the literature.

---

> ### Author Response · Authors · 2023-11-16
> **Response to Xinyi Wu**
>
> Dear Xinyi,
>
> Thanks for your valuable work. We did not know about your paper since your paper was accepted the same week we submitted our paper to ICLR.
>
> We found that you also revealed the weakness of the self-attention of GAT, which we agree with. We have already cited several papers about the Transformers' oversmoothing problem. We note that we are not the first who revealed the oversmoothing of Transformers but in fact, rely on those previous papers [1-8] to say that Transformers have the oversmoothing problem. As you may know, Transformers consist of self-attention, layer normalization, and feed-forward layers. Among them, most previous papers attribute the oversmoothing problem to the self-attention. We revised the sentence with the citation of your paper in Introduction as follows:
>
> In every self-attention layer, value vectors are aggregated in a weighted average manner since each row-wise sum of the attention matrix is always 1. Although each self-attention layer has its own attention matrix, this aggregation method causes the oversmoothing problem not only in Transformers but also in graph neural networks [1-15]. However, we confine our discussion to the oversmoothing of Transformers.
>
> > [1] Chengyue Gong, Dilin Wang, Meng Li, Vikas Chandra, and Qiang Liu. "Vision transformers with patch diversification." arXiv preprint arXiv:2104.12753, 2021.
> >
> > [2] Peihao Wang, Wenqing Zheng, Tianlong Chen, and Zhangyang Wang. "Anti-Oversmoothing in Deep Vision Transformers via the Fourier Domain Analysis: From Theory to Practice." ICLR, 2022.
> >
> > [3] Ameen Ali, Tomer Galanti, and Lior Wolf. "Centered Self-Attention Layers." arXiv preprint arXiv:2306.01610, 2023.
> >
> > [4] Han Shi, JIAHUI GAO, Hang Xu, Xiaodan Liang, Zhenguo Li, Lingpeng Kong, Stephen MS Lee, and James Kwok. "Revisiting Over-smoothing in BERT from the Perspective of Graph." ICLR, 2022.
> >
> > [5] Daquan Zhou, Bingyi Kang, Xiaojie Jin, Linjie Yang, Xiaochen Lian, Zihang Jiang, Qibin Hou, and Jiashi Feng. "Deepvit: Towards deeper vision transformer." arXiv preprint arXiv:2103.11886, 2021.
> >
> > [6] Hanqi Yan, Lin Gui, Wenjie Li, and Yulan He. "Addressing token uniformity in transformers via singular value transformation." UAI, 2022.
> >
> > [7] Yihe Dong, Jean-Baptiste Cordonnier, and Andreas Loukas. "Attention is not all you need: Pure attention loses rank doubly exponentially with depth." ICML, 2021.
> >
> > [8] Lorenzo Noci, Sotiris Anagnostidis, Luca Biggio, Antonio Orvieto, Sidak Pal Singh, and Aurelien Lucchi. "Signal propagation in transformers: Theoretical perspectives and the role of rank collapse." NeurIPS, 2022.
> >
> > [9] Xinyi Wu, Amir Ajorlou, Zihui Wu, and Ali Jadbabaie. "Demystifying Oversmoothing in Attention-Based Graph Neural Networks." NeurIPS, 2023.
> >
> > [10] Xinyi Wu, Zhengdao Chen, William Wei Wang, and Ali Jadbabaie. "A Non-Asymptotic Analysis of Oversmoothing in Graph Neural Networks." ICLR, 2023.
> >
> > [11] T. Konstantin Rusch, Michael M. Bronstein, and Siddhartha Mishra. "A survey on oversmoothing in graph neural networks." arXiv preprint arXiv:2303.10993, 2023.
> >
> > [12] Chen Cai and Yusu Wang. "A note on over-smoothing for graph neural networks." arXiv preprint arXiv:2006.13318, 2020.
> >
> > [13] Nicolas Keriven. "Not too little, not too much: a theoretical analysis of graph (over) smoothing.", NeurIPS, 2022.
> >
> > [14] Kenta Oono and Taiji Suzuki. "Graph Neural Networks Exponentially Lose Expressive Power for Node Classification." ICLR, 2020.
> >
> > [15] Guangtao Wang,  Rex Ying, Jing Huang, and Jure Leskovec. "Multi-hop attention graph neural network." IJCAI, 2021.

---

### Author Response · Authors · 2023-11-16
**Summary of Updated Paper**

We thank all reviewers for their constructive comments. We have uploaded the updated paper accordingly with the following major changes:
- Section 1: Update Introduction
- Section 2.2: Update background of graph convolutional filter
- Section 2.3: Update oversmoothing of Transformer
- Section 3.2: Update error bound of $E_K$
- Section 3.3: Update how to alleviate the oversmoothing problem
- Appendix A : Update the proof of Theorem 3.1
- Appendix B: Update the theoretical error bound in Figure 3
- Appendix H.2 : Add experimental results with standard deviations for graph classification task
- Appendix M: Add inference time for all experiments
- Appendix N: Add description for graph Fourier transform
- Appendix O: Add comparison for matrix polynomial vs. graph Fourier transform
- Appendix P: Add detail for first-order Taylor approximation of $\bar{A}^K$

---

### Author Response · Authors · 2023-11-16
**General Response to Reviewers (1/2)**

We thank all reviewers for their valuable time and insightful comments. We are appreciative and respectful of the fact that many reviewers are experts on Transformer and GNN. We make a few general remarks here and respond to individual comments to each review below.

---

**Q1. Reasons of Oversmoothing in Transformers**

In [1], it had already been revealed that the self-attention layer is analogous to the residual graph convolutional network [2]. In addition, there are several papers pointing out that the self-attention performs low-pass filterings [3,4]. We found that we cited them without much detail but just simply summarized their discoveries only. So, we revised our related work section to describe their detailed points (after removing unnecessary parts). We believe that it can help interested readers understand our motivation.

> [1] Han Shi, JIAHUI GAO, Hang Xu, Xiaodan Liang, Zhenguo Li, Lingpeng Kong, Stephen MS Lee, and James Kwok. "Revisiting Over-smoothing in BERT from the Perspective of Graph." In International Conference on Learning Representations. 2022.
>
> [2] Thomas N. Kipf, and Max Welling. "Semi-Supervised Classification with Graph Convolutional Networks." In International Conference on Learning Representations. 2017.
>
> [3] Peihao Wang, Wenqing Zheng, Tianlong Chen, and Zhangyang Wang. "Anti-Oversmoothing in Deep Vision Transformers via the Fourier Domain Analysis: From Theory to Practice." In International Conference on Learning Representations. 2022.
>
> [4] Jiawang Bai, Li Yuan, Shu-Tao Xia, Shuicheng Yan, Zhifeng Li, and Wei Liu. "Improving vision transformers by revisiting high-frequency components." In European Conference on Computer Vision, pp. 1-18. Cham: Springer Nature Switzerland, 2022.

The public comment by Xinyi Wu is also about this question. So, we refer other interested reviewers to [our response to Xinyi](https://openreview.net/forum?id=poFAoivHQk&noteId=kfAeVcTpOU).

---

**Q2. Diagonalizability of Adjacency Matrix**
We found that our unnecessary descriptions on graph filtering in our related work section caused this misunderstanding. There are two paradigms of implementing graph filters: i) $\textcolor{blue}{\text{matrix polynomial}}$ which does not require diagonalizability, ii) $\textcolor{red}{\text{graph Fourier transform}}$ which uses the eigendecomposition for diagonalizable adjacency matrices or the Jordan decomposition for non-diagonalizable adjacency matrices [5]. In the following equation, the blue one is the matrix polynomial-based graph filter and the red one is the graph Fourier transform-based graph filter after the spectral decomposition. $\mathbf{y}$ means the filtered signal.

$$\mathbf{y} ={\color{blue}{\sum_{k=0}^{K} w_k \mathbf{S}^k\mathbf{x}}} = \sum_{k=0}^{K} \mathbf{V}^\intercal w_k\mathbf{\Lambda}^k \mathbf{V}\mathbf{x}  = \mathbf{V}^\intercal \big(\sum_{k=0}^{K}  w_k\mathbf{\Lambda}^k \big) \mathbf{V}\mathbf{x} = {\color{red}{\mathbf{V}^\intercal g(\mathbf{\Lambda}) \mathbf{V}\mathbf{x}}}$$

Those two paradigms have their own weaknesses: i) the matrix polynomial approach requires explicit matrix multiplications, and ii) the graph Fourier transform approach requires expansive spectral decompositions. The matrix polynomial is preferred when the number of matrix multiplications is not many. Otherwise, the graph Fourier transform approach may be better since the matrix multiplication can be simplified after the eigendecomposition or the Jordan decomposition (or the Jordan normal form). We refer to an Internet blog for this: https://math.stackexchange.com/questions/2351632/why-do-we-need-a-jordan-normal-form.

Among those two, we use the first matrix polynomial approach with only three non-zero coefficients $\\{w_0, w_1, w_K\\}$ since it does not require the complicated spectral decomposition. However, we explained the graph Fourier transform by diagonalizing adjacency matrices in the related work section of our initial submission, which caused all your misunderstanding. We apologize for this. So, we completely removed it from the main content and maintain only the matrix polynomial-based description. Content removed from the main text has been moved to Appendix N. Since we do not rely on any explicit spectral decomposition but on the matrix polynomial, any adjacency matrix can be used. As you may know, the adjacency matrix is a representative shift operator for a given graph.

One can argue that graph filters created only by three non-zero coefficients $\\{w_0, w_1, w_K\\}$ are simple. However, this is not true since many other recent GCN methods, e.g., GPR-GNN, GREAD, and so on, are special cases of our method (see our answer for [Q3.](https://openreview.net/forum?id=poFAoivHQk&noteId=ozboZFBU0m) in general response (2/2)).

> [5] Aliaksei Sandryhaila, and José MF Moura. "Discrete signal processing on graphs." IEEE Transactions on signal processing 61, no. 7 (2013): 1644-1656.

---

> ### Author Response · Authors · 2023-11-16
> **General Response to Reviewers (2/2)**
>
> **Q3. How does our filter prevent the oversmoothing and why do we need $\bar{\mathbf{A}}^2 - \bar{\mathbf{A}}$?**
>
> The key leading to the behaviors of our proposed filter is the coefficients $\\{w_0, w_1, w_K\\}$ --- note that in the self-attention of Transformers, $w_0 = w_K = 0$ and $w_1 = 1$. Since our method can learn any appropriate values for them for a downstream task, it can be reduced to low-pass-only, high-pass-only, or combined filters.
>
> In particular, our filter reduces to GPR-GNN and GREAD, two popular methods preventing the oversmoothing for graph convolutional networks, in certain learned settings on $\\{w_0, w_1, w_K\\}$ (see our discussion in Section 3.3 in our revised paper).
>
> We use $\bar{\mathbf{A}}^2 - \bar{\mathbf{A}}$ to approximate $\bar{\mathbf{A}}^K$ (see Eq. (4) in our revised paper). As mentioned earlier, the low/high-pass filter characteristics are decided by $\{w_0, w_1, w_K\}$. Therefore, the approximated $\bar{\mathbf{A}}^K$ by $(K-1)(\bar{\mathbf{A}}^2 - \bar{\mathbf{A}})$ and the coefficients $\{w_0, w_1, w_K\}$ can alleviate the oversmoothing problem of the self-attention by constituting an appropriate graph filter for a downstream task.
>
> ---
>
> **Q4. What are the targets of our graph filtering?**
>
> We reiterate that we do not explicitly project node features onto the eigenspace of $\bar{\mathbf{A}}$ but use the matrix polynomial. We note that the self-attention of Transformers calculates the weighted average of the value vectors of tokens. Our method also applied our designed graph filter to the value vectors of tokens. However, our method is able to extract high-frequency information from the value vectors, which is different from the self-attention, which can remove only the low-frequency information, i.e., the weighted average.

---

### Comment · Reviewer_kKdF · 2023-11-17
**Theoretical Concerns From Directed Graph Signal Processing**

Dear reviewers and area chairs:

I am writing to present an assessment of this paper which introduces a novel approach termed Graph Filter-based Self-Attention (GFSA), predicated on what the authors refer to as a matrix polynomial methodology. This paper considers the attention matrix of Self-Attention as a directed graph. Drawing upon my extensive understanding of Directed Graph Signal Processing (DGSP), it is commonly acknowledged that the preferred method for addressing a directed graph involves the transformation of the Directed Graph Signal Operator (DGSO) into a diagonalizable form, particularly when the DGSO is non-diagonalizable [1-11]. This process entails incorporating the diagonalizable DGSO into the graph filter, underpinned by the rationale that each signal space – corresponding to individual columns of input features – should maintain its inherent characteristics post-filtering. In simpler terms, the signal space encompassing each column of the output vectors should align seamlessly with that of the input vectors. This preservation of signal space is contingent upon the diagonalizability of the DGSO [2,3,4].

The authors, however, elect to bypass this critical preprocessing stage of transforming the DGSO into a diagonalizable form. Instead, they directly implement the non-transformed DGSO in a filter, asserting that the resulting GFSA still qualifies as a graph convolution. It is imperative to note that such a procedure should more accurately be described as a directed graph convolution. While this methodology does demonstrate commendable performance, it fundamentally challenges and potentially disrupts the established theoretical framework of DGSP. This divergence from conventional theory leaves me, and potentially others in the field, unable to satisfactorily reconcile the authors' approach with existing DGSP principles.

Given the potential impact and the likely extensive citations this paper could garner across both Graph Neural Networks (GNNs) and DGSP domains if accepted, I feel compelled, in my capacity as a reviewer for ICLR 2024, to urge my fellow reviewers and area chairs to approach the evaluation of this submission with heightened scrutiny and consideration. The implications of endorsing a paper that deviates so significantly from accepted theoretical underpinnings should be weighed carefully, to ensure the integrity and forward progression of our field.

#### Reference
> [1] Marques, A. G., Segarra, S., & Mateos, G. (2020). Signal Processing on Directed Graphs: The Role of Edge Directionality When Processing and Learning From Network Data. IEEE Signal Processing Magazine, 37(6), 99–116.
>
> [2] Sandryhaila, A., & Moura, J. M. F. (2013). Discrete signal processing on graphs: Graph fourier transform. 2013 IEEE International Conference on Acoustics, Speech and Signal Processing, 6167–6170.
>
> [3] Sandryhaila, A., & Moura, J. M. F. (2013). Discrete Signal Processing on Graphs. IEEE Transactions on Signal Processing, 61(7), 1644–1656.
>
> [4] Sandryhaila, A., & Moura, J. M. F. (2014). Discrete Signal Processing on Graphs: Frequency Analysis. IEEE Transactions on Signal Processing, 62(12), 3042–3054.
>
> [5] Singh, R., Chakraborty, A., & Manoj, B. S. (2016). Graph Fourier transform based on directed Laplacian. 2016 International Conference on Signal Processing and Communications (SPCOM), 1–5.
>
> [6] Deri, J. A., & Moura, J. M. F. (2017). Spectral Projector-Based Graph Fourier Transforms. IEEE Journal of Selected Topics in Signal Processing, 11(6), 785–795.
>
> [7] Domingos, J., & Moura, J. M. F. (2020). Graph Fourier Transform: A Stable Approximation. IEEE Transactions on Signal Processing, 68, 4422–4437.
>
> [8] Barrufet, J., & Ortega, A. (2021). Orthogonal Transforms for Signals on Directed Graphs. arXiv Preprint arXiv: Arxiv-2110. 08364.
>
> [9] Chung, F. (04 2005). Laplacians and the Cheeger Inequality for Directed Graphs. Annals of Combinatorics, 9(1), 1–19.
>
> [10] Fanuel, M., Alaíz, C. M., & Suykens, J. A. K. (02 2017). Magnetic eigenmaps for community detection in directed networks. Phys. Rev. E, 95, 022302.
>
> [11] Fanuel, M., Alaíz, C. M., Fernández, Á., & Suykens, J. A. K. (2018). Magnetic Eigenmaps for the visualization of directed networks. Applied and Computational Harmonic Analysis, 44(1), 189–199.

---

### Author Response · Authors · 2023-11-18
**Summary of our paper**

Dear all reviewers and area chairs,

Let me summarize the background theory and our contributions again.

**1. Theoretical background**

Let me start with the following famous derivation.

> $$\mathbf{y} = \mathbf{H}\mathbf{x} = {\color{blue}{\sum_{k=0}^{K} w_k \mathbf{A}^k\mathbf{x}}} = \sum_{k=0}^{K} \mathbf{V}^\intercal w_k\mathbf{\Lambda}^k \mathbf{V}\mathbf{x}  = {\color{red}{\mathbf{V}^\intercal \big(\sum_{k=0}^{K}  w_k\mathbf{\Lambda}^k \big) \mathbf{V}\mathbf{x}}},$$where the blue method is called $\textit{matrix-polynomial}$ and the red method is called $\textit{spectral-decomposition}$. Let $\mathbf{x}$ be a graph signal, i.e., node feature, and $\mathbf{H}$ be a graph filter. $\mathbf{A}$ means a graph shift operator, e.g., adjacency matrix.

$\mathbf{H}$ can be mathematically any filter, but we typically confine it to $\textit{lienar}$ and $\textit{shift-invariant}$ forms. In such cases, $\mathbf{H}\mathbf{x} = {\color{blue}{\sum_{k=0}^{K} w_k \mathbf{A}^k\mathbf{x}}}$ is true [1, Theorem 1]. For other complicated forms of $\mathbf{H}$, we typically do not study since they are hard to deal with.

However, there is a problem in $\mathbf{H}\mathbf{x} = {\color{blue}{\sum_{k=0}^{K} w_k \mathbf{A}^k\mathbf{x}}}$. $K$ must be the same as the number of nodes for the matrix polynomial to exactly reproduce $\mathbf{H}$. As you may know, calculating $\mathbf{A}^{N-1}$, where $N$ is the number of nodes, is computationally prohibitive for large graphs.

We then rely on the spectral decomposition since ${\color{blue}{\sum_{k=0}^{K} w_k \mathbf{A}^k\mathbf{x}}} = {\color{red}{\mathbf{V}^\intercal \big(\sum_{k=0}^{K}  w_k\mathbf{\Lambda}^k \big) \mathbf{V}\mathbf{x}}}$. After the spectral decomposition, the high-order matrix powers are changed to the element-wise scalar powers of eigenvalues.

For scientific domains, exact computations are important. Therefore, ones typically use the spectral decomposition method since after the spectral decomposition, remaining calculations become trivial.

For deep learning, ones do not explicitly perform the spectral decomposition since it is computationally prohibitive (sometimes ones perform an iterative spectral decomposition algorithm to know only low-frequency components though if needed). Instead, ones set $k$ to a small value. In such a case, of cource, $\mathbf{H}\mathbf{x} = {\color{blue}{\sum_{k=0}^{K} w_k \mathbf{A}^k\mathbf{x}}}$ is not true any more since $K = N$ is needed to be exact. We should learn $w_k$ for all $k$.

However, utilizing lower-order polynomial terms only is a good trade-off between computational efficiency and graph filter efficacy for deep learning. In GCN by Kipf and Welling [2], they explicitly said that they avoid the spectral decomposition by using the lowest-order terms of the matrix polynomial. They set $K=1$. Therefore, what they are doing is conceptually similar to $\mathbf{H}\mathbf{x} \approx {\color{blue}{\sum_{k=0}^{1} w_k \mathbf{A}^k\mathbf{x}}}$ (we omit other details for brevity and our simple discussion). Except GCN by Kipf and Welling, there are many other papers utilizing the matrix polynomial approach, e.g., ChebNet [3]. I can list a long sequence of papers for this, but for brevity, I will not do it.

Reviewer kKdf thinks in the reverse order that $\mathbf{y} = {\color{red}{\mathbf{V}^\intercal \big(\sum_{k=0}^{K}  w_k\mathbf{\Lambda}^k \big) \mathbf{V}\mathbf{x}}}$ is the first invented, and later ${\color{red}{\mathbf{V}^\intercal \big(\sum_{k=0}^{K}  w_k\mathbf{\Lambda}^k \big) \mathbf{V}\mathbf{x}}} = {\color{blue}{\sum_{k=0}^{K} w_k \mathbf{A}^k\mathbf{x}}}$ follows. However, this is unfortunately a misconception. The correct order is the equation that I wrote above.

Therefore, the key point is that for $\mathbf{H}\mathbf{x} = {\color{blue}{\sum_{k=0}^{K} w_k \mathbf{A}^k\mathbf{x}}}$, we require only $\textit{lienar}$ and $\textit{shift-invariant}$ forms. The $\textit{diagonalizability}$ is required only when ${\color{blue}{\sum_{k=0}^{K} w_k \mathbf{A}^k\mathbf{x}}} = {\color{red}{\mathbf{V}^\intercal \big(\sum_{k=0}^{K}  w_k\mathbf{\Lambda}^k \big) \mathbf{V}\mathbf{x}}}$ is used.

**2. Contributions**

We consider the self-attention matrix as directed normalized adjacency matrix, which was already exploited several times by some other works [4-10].  We are not the first saying that the self-attention matrix is basically directed normalized adjacency matrix. Note that the row-wise sum of the self-attention matrix is 1 due to the softmax and it contains weights among tokens' value vectors. Therefore, this interpretation is natural to understand.

We want to construct an effective filter with a higher-order polynomial term. However, this is computationally demanding. Therefore, we proposed a method to approximate  $\mathbf{A}^K$ with $\mathbf{A}^2 - \mathbf{A}$. We applied our filter to almost all domains where Transformers show good performance and achieve non-trivial enhancements.

---

> ### Author Response · Authors · 2023-11-18
> **References & Final Remark**
>
> For your convenience, I list the direct pdf links below.
>
> [1] https://users.ece.cmu.edu/~asandryh/papers/icassp13.pdf
>
> [2] https://arxiv.org/pdf/1609.02907.pdf
>
> [3] https://arxiv.org/pdf/1606.09375.pdf
>
> [4] https://arxiv.org/abs/2303.06562
>
> [5] https://arxiv.org/abs/2203.05962
>
> [6] https://arxiv.org/abs/2306.01610
>
> [7] https://arxiv.org/abs/2202.08625
>
> [8] https://arxiv.org/abs/2110.09443
>
> [9] https://arxiv.org/abs/2110.09443
>
> [10] https://arxiv.org/abs/2211.14208
>
> We thank all the reviewers for the discussion. While the discussion regarding matrix polynomials is valid, it remains our opinion that the deviation from classical spectral decomposition theory is well worth the drastic improvement in performance. There is substantial evidence in the literature that our stance is not unorthodox as I showed in my previous message. We will add a remark to the paper acknowledging the theoretical consequences of the choice. It is our opinion that there is value in a diversity of research that does not always align with older theory, and we have put substantial effort into developing careful and rigorous benchmarks so that the value of our treatment of the matrix polynomials is scientifically supported. The performance gains we have realized are substantial, and we believe it to be inappropriate to suppress sharing these advances with the community.
>
> Sincerely,
>
> Authors

---

### Comment · Reviewer_kKdF · 2023-11-19
**Summry of the authors' misunderstandings**

The origin of GNNs is generally accepted to be the Graph Convolutional Network (GCN) [1]. Previously, similar neural networks existed but were known as CNNs on graphs [2][3]. GCNs primarily study undirected graphs, characterized by symmetric adjacency matrices. In linear algebra, symmetric matrices are diagonalizable. This principle is foundational in subsequent studies like SGC [4], GDC [5], APPNP [6], SSGC [7], and GPR-GNN [8], all of which focus on undirected graphs. This approach is in line with graph signal processing, where the self-looped normalized adjacency matrix of an undirected graph is used in polynomial-based filters to boost a model's expressiveness.

However, when it comes to directed graphs, the scenario changes. Directed graphs have asymmetric adjacency matrices, which are non-diagonalizable. In directed graph signal processing, using such an adjacency matrix directly as a shift operator in a polynomial-based filter does not filter the eigenvalues of the adjacency matrix. Prior to this paper, research on GNNs for directed graphs, like MagNet [9], employed the magnetic adjacency matrix with a self-loop as the directed graph shift operator (DGSO) in Chebyshev polynomials of the first kind, essentially using a diagonalizable DGSO.

The authors persist in refusing to differentiate between undirected and directed graphs, as well as between GNNs that process these two types of graphs. Models like GCN, SGC, GDC, APPNP, SSGC, and GPR-GNN, designed for undirected graphs, incorporate a self-looped normalized adjacency matrix into a polynomial-based filter, consistent with graph signal processing principles. However, for directed graphs, these models are inapplicable. This has led to the development of GNNs specifically for directed graphs, like DiGCN [10], MagNet, and Signed Graph Neural Networks [11], which use a diagonalizable adjacency matrix of a directed graph with self-loop.

Furthermore, in this review, I never wrote that $y = \mathbf{V}\left(\sum\_{k=0}^{K}w_{k}\mathbf{\Lambda}^{k}\right)\mathbf{V}^{-1}\mathbf{x}$ was the first invented. The authors misrepresented my meaning. I dispute the notion that the Graph Fourier Transform (GFT) predates polynomial-based graph filters. The relationship between the two is synergistic rather than hierarchical. It's crucial to clarify that GFT lays the groundwork for understanding polynomial-based graph filters, rather than being a direct method of implementation. Disappointingly, the author remains confined to the theoretical framework of graph signal processing in their proposed GFSA, inaccurately using GNNs for undirected graphs as examples.

In conclusion, I strongly advise the authors to recognize the distinct differences between undirected and directed graphs, the GNNs that process each type, and the separate fields of graph signal processing and directed graph signal processing.

---

> ### Comment · Reviewer_kKdF · 2023-11-19
>
> ### Reference
> [1] Kipf, T. N., & Welling, M. (2017). Semi-Supervised Classification with Graph Convolutional Networks. International Conference on Learning Representations.
>
> [2] Bruna, J., Zaremba, W., Szlam, A., & LeCun, Y. (2014). Spectral Networks and Locally Connected Networks on Graphs. In Y. Bengio & Y. LeCun (Eds.), International Conference on Learning Representations.
>
> [3] Defferrard, M., Bresson, X., & Vandergheynst, P. (2016). Convolutional Neural Networks on Graphs with Fast Localized Spectral Filtering. In D. Lee, M. Sugiyama, U. Luxburg, I. Guyon, & R. Garnett (Eds.), Advances in Neural Information Processing Systems (Vol. 29). Curran Associates, Inc.
>
> [4] Wu, F., Souza, A., Zhang, T., Fifty, C., Yu, T., & Weinberger, K. (06 2019). Simplifying Graph Convolutional Networks. In K. Chaudhuri & R. Salakhutdinov (Eds.), Proceedings of the 36th International Conference on Machine Learning (pp. 6861–6871). PMLR.
>
> [5] Gasteiger, J., Weiß enberger, S., & Günnemann, S. (2019). Diffusion Improves Graph Learning. In H. Wallach, H. Larochelle, A. Beygelzimer, F. d\textquotesingle Alché-Buc, E. Fox, & R. Garnett (Eds.), Advances in Neural Information Processing Systems (Vol. 32). Curran Associates, Inc.
>
> [6] Gasteiger, J., Bojchevski, A., & Günnemann, S. (2019). Combining Neural Networks with Personalized PageRank for Classification on Graphs. International Conference on Learning Representations.
>
> [7] Zhu, H., & Koniusz, P. (2021). Simple Spectral Graph Convolution. International Conference on Learning Representations.
>
> [8] Chien, E., Peng, J., Li, P., & Milenkovic, O. (2021). Adaptive Universal Generalized PageRank Graph Neural Network. International Conference on Learning Representations.
>
> [9] Zhang, X., He, Y., Brugnone, N., Perlmutter, M., & Hirn, M. (2021). MagNet: A Neural Network for Directed Graphs. In A. Beygelzimer, Y. Dauphin, P. Liang, & J. W. Vaughan (Eds.), Advances in Neural Information Processing Systems.
>
> [10] Tong, Z., Liang, Y., Sun, C., Li, X., Rosenblum, D., & Lim, A. (2020). Digraph Inception Convolutional Networks. In H. Larochelle, M. Ranzato, R. Hadsell, M. F. Balcan, & H. Lin (Eds.), Advances in Neural Information Processing Systems (Vol. 33, pp. 17907–17918). Curran Associates, Inc.
>
> [11] Singh, R., & Chen, Y. (2023). Signed Graph Neural Networks: A Frequency Perspective. Transactions on Machine Learning Research.

---

### Author Response · Authors · 2023-11-19
**linear, shift-invariant vs. linear, shift-invariant, diagonalizable**

Dear All,

Let me simplify our discussion. Given the following equation,

> $$\mathbf{y} = \mathbf{H}\mathbf{x} = {\color{blue}{\sum_{k=0}^{K} w_k \mathbf{A}^k\mathbf{x}}} = \sum_{k=0}^{K} \mathbf{V}^\intercal w_k\mathbf{\Lambda}^k \mathbf{V}\mathbf{x}  = {\color{red}{\mathbf{V}^\intercal \big(\sum_{k=0}^{K}  w_k\mathbf{\Lambda}^k \big) \mathbf{V}\mathbf{x}}},$$where the blue method is called $\textit{matrix-polynomial}$ and the red method is called $\textit{spectral-decomposition}$. Let $\mathbf{x}$ be a graph signal, i.e., node feature, and $\mathbf{H}$ be a graph filter. $\mathbf{A}$ means a graph shift operator, e.g., adjacency matrix.

**1. Linear, Shift-Invariant, and Diagonalizable**

This is the ideal case that Reviewer kKdF says, and many GNNs deal with this setting. In fact, it is true that many engineers feel comfortable since they may be the most familiar with this setting. In this situation, we use the spectral-decomposition and do not need to perform the matrix power since $\mathbf{y} = \mathbf{H}\mathbf{x} = {\color{red}{\mathbf{V}^\intercal \big(\sum_{k=0}^{K}  w_k\mathbf{\Lambda}^k \big) \mathbf{V}\mathbf{x}}}$. Even in this setting, however, many GNNs still use the matrix polynomial only with low-order polynomial terms due to the high complexity of the spectral decomposition.


**2. Linear and Shift-Invariant**

This is the setting that our paper uses. In this situation, it is guaranteed that $\mathbf{y} = \mathbf{H}\mathbf{x} = {\color{blue}{\sum_{k=0}^{K} w_k \mathbf{A}^k\mathbf{x}}}$ [1, Theorem 1].

I copy the following description from [2]. In [2], Eq. (4) means the matrix polynomial method and Eq. (3) means the spectral-decomposition method. This description says that we can still use the matrix polynomial method even when non-diagonalizable since Eq. (4) still holds. GSO means graph shift operator, such as, adjacency matrix.

>If the GSO is not diagonalizable, the generalization of (3) is unclear, while (4) still holds.

In general, this paper [2] shows that directed adjacency matrix can be used for graph signal processing.

**3. Graph filters**

Given $\mathbf{y} = \mathbf{H}\mathbf{x}$, as a matter of fact, any $\mathbf{H}$ is mathematically a graph filter when it converts $\mathbf{x}$ to $\mathbf{y}$. Of course, many such filters are meaningless in practice. In practice, we are interested in denoising filters, blurring filters, etc. Fortunately, many such meaningful filters satisfy the requirement of (linear), (linear, shift-invariant) or (linear, shift-invariant, diagonalizable).

As you know, classical discrete signal processing is a special case of graph signal processing, where directed ring graphs are used [1, Figure 1(a)]. Kalman filter is a representative linear but non-shift-invariant filter. In various domains across engineering and scientific areas, it is not true that only linear, shift-invariant, and diagonalizable filters are used.

I respect Reviewer kKdF for its conservative perspective that we need to be careful when designing a method, but our design is also well within the theoretical boundary. Let me conclude at this moment and leave further discussion to reviewers and chairs.

[1] https://users.ece.cmu.edu/~asandryh/papers/icassp13.pdf

[2] https://www.hajim.rochester.edu/ece/sites/gmateos/pubs/dgft/DGFT_SPMAG.pdf

Sincerely,

Authors

---

> ### Author Response · Authors · 2023-11-22
> **Additional Justification**
>
> Let us compare the self-attention matrix with the symmetrically normalized adjacency matrix [1]. We note that the symmetrically normalized adjacency matrix does not mean the the symmetric normalized adjacency matrix. In [1], they define the following symmetrically normalized adjacency matrix (SNA) for directed graphs:
> $$\mathbf{D}\_{in}^{-1/2} \mathbf{A} \mathbf{D}_{out}^{-1/2}.$$
>
> Since the self-attention matrix $\bar{\mathbf{A}}$'s row-wise sum is always 1 after softmax, the following is the case, where $N$ is the number of tokens:
>
> $$\bar{\mathbf{A}} = \mathbf{D}^{-1}\mathbf{A}=\frac{1}{N}\mathbf{A}.$$
>
>
> Since the degree of every node is $N$ in the self-attention matrix, the following is the case:
>
> $$\mathbf{D}\_{in}^{-1/2}\mathbf{A}\mathbf{D}_{out}^{-1/2}=\mathbf{D}^{-1/2}\mathbf{A}\mathbf{D}^{-1/2}=\frac{1}{\sqrt{N}}\mathbf{A}\frac{1}{\sqrt{N}}=\frac{1}{N}\mathbf{A}.$$
>
> Therefore, the self-attention matrix is a special case of SNAs. In [1], they propose to use the singular value domain analysis which provides a similar notion of analysis to the frequency domain analysis. As you may know the singular value decomposition is a sort of relaxed (or generalized) version of the eigendecomposition, where $\mathbf{A} = \mathbf{U}\boldsymbol{\Sigma}\mathbf{V}^\intercal$ in the singular value decomposition --- note that in the eigendecomposition, $\mathbf{U} = \mathbf{V}$. For the singular value domain analysis, we analyze the frequency response of $\boldsymbol{\Sigma}$ to understand the high/low-pass characteristics of filters. In addition, the following paper [2] also uses the singular value domain analysis for directed graphs.
>
> In Fig. 2, we have already reported the singular value domain analysis results with and without our proposed GFSA.
>
> We were reluctant to strongly argue the connection to SNA, but while preparing for the rebuttal, we have recognized that the degree of every node is $N$ in the self-attention. Thus, we modified our paper in blue regarding the symmetrically normalized adjacency matrix after citing [1] and rely on the method suggested in [1] which is theoretically sound. Please consider this point for your evaluation.
>
> **Therefore, we rely on the matrix polynomial to learn graph filters (see our recent global response whose title is "linear, shift-invariant vs. linear, shift-invariant, diagonalizable") and for the frequency response analysis, rely on the singular value domain analysis.**
>
>
> [1] Sohir Maskey, Raffaele Paolino, Aras Bacho, and Gitta Kutyniok, "A Fractional Graph Laplacian Approach to Oversmoothing," In NeurIPS, 2023, https://arxiv.org/pdf/2305.13084.pdf
>
> [2] Zou, Chunya, Andi Han, Lequan Lin, and Junbin Gao, "A simple yet effective SVD-GCN for directed graphs," arXiv preprint arXiv:2205.09335, 2022, https://arxiv.org/pdf/2205.09335.pdf

---

> ### Comment · Reviewer_kKdF · 2023-11-22
>
> Thank you for providing your justification. It helps in understanding your perspective and approach. I am very surprised by the paper [1]. Ultimately, I decided to increase the score.
>
> [1] Maskey, S., Paolino, R., Bacho, A., & Kutyniok, G. (2023). A Fractional Graph Laplacian Approach to Oversmoothing. Thirty-Seventh Conference on Neural Information Processing Systems.

---

### Comment · Reviewer_fbar · 2023-11-22
**Strong support for the authors**

Hi all,

Having followed the discussion in the last week, I would like to maintain my high score for this paper.  Given that this article kept high quality at the submission, this very discussion will have sharpened the scientific presentation and made the authors scrutinize the mathematical underpinnings and presentation of their work, increasing the value of the work further ("We will add a remark to the paper acknowledging the theoretical consequences of the choice.", as the authors have said in this discussion, which is positive).

I would like to cite another part of the authors' many thorough replies during the rebuttal period, namely the following:  "It is our opinion that there is value in a diversity of research that does not always align with older theory, and we have put substantial effort into developing careful and rigorous benchmarks so that the value of our treatment of the matrix polynomials is scientifically supported. The performance gains we have realized are substantial, and we believe it to be inappropriate to suppress sharing these advances with the community."

Although this discussion ultimately most likely was a scientifically positive thing for the authors -- being forced to think about their assumptions in such detail, in my view it slightly discredits reviewer kKdF that they [reviewer kKdF] were so thoroughly skeptic, and not acknowledging any other positive aspects of the paper other than the experimental. Anybody who has written a machine learning paper will know that a more than substantial amount of work went into a paper like this one (it is a very ambitious paper), from the conceptualization, theoretical motivation and into the experimental phase, and as a reviewer I find it important to at least acknowledge this or encourage the work in progress and not simply frown at something because it doesn't follow the exact theoretical school the reviewer comes from. I find the tone in many of kKdF's comments being slightly dogmatist or querulant, which is where I find the author's above cited comment important to keep in mind. But at the same time, it is fantastic that kKdF took the time to go into precisely these details, which they know better than me, so we ended up with the high quality final paper that we have here. So thank you to both kKdF and the authors, in the end.

This article is greatly interesting for anyone interested in the expressivity of Transformers (which is a large number of researchers at this point), and how their limits can be pushed. It is clearly presented with thorough experimentation and as far as I can see is based on uncontroversial matrix algebraical explanations. In my view, then, whether it can 'truly' say that it adheres to the school of graph signal processing or not matters less.

The authors highly thorough responsiveness through the rebuttal period has convinced me even more about my score. Thus I vote for accepting this paper.

---

> ### Comment · Reviewer_kKdF · 2023-11-22
>
> I am very upset that Reviewer fbar described me as 'were so thoroughly skeptic, and not acknowledging any other positive aspects of the paper other than the experimental', just because I questioned the author's mistakes from a professional graph signal processing perspective. As a reviewer for ICLR 2024, it is my responsibility to question the author's oversights from a professional perspective. Furthermore, I am very grateful for the defense presented by the authors in this rebuttal.

---

> ### Author Response · Authors · 2023-11-22
> **Thanks for your support.**
>
> We greatly appreciate the reviewer fbar in acknowledging the contributions we made in this paper. We also appreciate the reviewer kKdF for providing their perspective, which we were less careful about properly articulating the difference by the time we initially work on this project.
>
> After all, we appreciate all these conversations with the reviewers; we share the same research interest and attempt to make advancements on the topic in a scientifically proper way. Our hope is that the reviews kKdF and huFG could acknowledge that the contributions we made in this paper is nontrivial. Our method design is well within the graph signal processing theory (although it is little different from the most popular graph signal processing setting) and its empirical evidence well justifies it.
>
> We hope that our research can contribute to our community since all of us, as researchers, should use our knowledge and experience for the technological advancement of mankind.
>
> Sincerely,
>
> Authors

---

> ### Comment · Reviewer_kKdF · 2023-11-23
>
> Thank you for analyzing GFSA from the perspective of singular values, rather than through graph signal processing. This approach is helpful in breaking the boundary between the directed graph convolution and the self-attention, which I am very pleased to see.

---

### Meta-Review · Area_Chair_iQ8u · 2023-12-12

**Metareview:**

I appreciate the efforts from both the authors and reviewers in the discussion period, especially Reviewer kKdF and fbar. Given the polarized ratings and heated discussions, I carefully read both the post-rebuttal version of the paper (including some of the appendix) and the reviews before making my recommendations.

A main focus of the discussion lies on the technical justification of the proposed method. While I agree with Reviewer fbar that the technical justification does not need to adhere to the school of graph signal processing, with an independent examination, I find that this paper has significant problems undermining its technical soundness.

- One major problem is theorem 3.1, which is the main theoretical result justifying the proposed method. Since the L-infinity norm of A^K is 1, an all-zero matrix has an approximation error of 1. This renders the error bound 2K in theorem 3.1 almost meaningless.
- The approximated derivative in Equation (4) is a key step leading to the proposed method. However, this step is not sufficiently justified. A key approximation step provided in Appendix P referred to Brouwer et al. (2019). It's unclear from the text of Appendix P how Brouwer et al. (2019) connect to the proposed method. A brief search of keywords such as "Taylor approximation" and or even "approximation" in the referred paper does not lead to anything insightful either.
- A minor problem with the definition of discrete convolution in Equation (2): g is an n-dim vector and some i-j indices are negative so g_{i-j} will be undefined. This can be easily fixed though.

In terms of other aspects,

- Clearly the extensive experiments and empirical gain of the proposed method is an important strength of this paper. However, I find that the empirical gain is often marginal on most of the experimental settings. I definitely do not want to penalize the authors by disclosing more results---I appreciate it---but Figure 1 of this paper does seem to be a bit over-claiming/misleading given the more detailed results.
- In terms of novelty and significance, I agree with Reviewer huFG that the connection between self-attention and weighted graphs has been extensively discussed and the significance of the new graph filter perspective in comparison to those prior works is unclear. The author response to this concern essentially reiterates that the perspective of graph filter is new, which fails to address this concern about significance.

Overall, while this paper has surely conducted an impressive set of experiments, proper technical justification of the proposed method is largely lacking. Additionally, the empirical gain, with a more detailed look, isn't as significant as Figure 1 suggests.

Finally, I'd like to make a rather subjective comment regarding the criterion of publication. IMHO, while not every method should have strong theoretical justification to be published, a due diligence for technical correctness of everything written in the paper is necessary for publication. The preliminary ideas and empirical results can still be frictionlessly disseminated to the community (through platforms such as arxiv or openreview) before having everything correct. But we as reviewers should be gatekeepers for the technical soundness (even more than novelty and significance) of the papers. This is also the core spirit of the recently found Transaction of Machine Learning Research (TMLR).

**Justification For Why Not Higher Score:**

Problems with technical soundness. Vague significance.

**Justification For Why Not Lower Score:**

N/A

---

### Decision · Program_Chairs · 2024-01-16

Reject